# Post mortem evaluation of inflammation, oxidative stress, and PPARγ activation in a nonhuman primate model of cardiac sympathetic neurodegeneration

Jeanette M. Metzger[1,2], Helen N. Matsoff[1], Alexandra D. Zinnen[1], Rachel A. Fleddermann[1¤], Viktoriya Bondarenko[1], Heather A. Simmons[1], Andres Mejia[1], Colleen F. Moore[3], Marina E. Emborg[1,2,4]*

1 Wisconsin National Primate Research Center, University of Wisconsin–Madison, Madison, WI, United States of America, 2 Cellular and Molecular Pathology Graduate Program, University of Wisconsin–Madison, Madison, WI, United States of America, 3 Department of Psychology, University of Wisconsin–Madison, Madison, WI, United States of America, 4 Department of Medical Physics, University of Wisconsin–Madison, Madison, WI, United States of America

¤ Current address: School of Medicine, The University of New Mexico, Albuquerque, NM, United States of America
* emborg@primate.wisc.edu

**Data Availability Statement:** All relevant data are within the paper and its Supporting Information files.

## Abstract

Cardiac dysautonomia is a common nonmotor symptom of Parkinson's disease (PD) associated with loss of sympathetic innervation to the heart and decreased plasma catecholamines. Disease-modifying strategies for PD cardiac neurodegeneration are not available, and biomarkers of target engagement are lacking. Systemic administration of the catecholaminergic neurotoxin 6-hydroxydopamine (6-OHDA) recapitulates PD cardiac dysautonomia pathology. We recently used positron emission tomography (PET) to visualize and quantify cardiac sympathetic innervation, oxidative stress, and inflammation in adult male rhesus macaques (*Macaca mulatta;* n = 10) challenged with 6-OHDA (50mg/kg; i.v.). Twenty-four hours post-intoxication, the animals were blindly and randomly assigned to receive daily doses of the peroxisome proliferator-activated receptor gamma (PPARγ) agonist pioglitazone (n = 5; 5mg/kg p.o.) or placebo (n = 5). Quantification of PET radioligand uptake showed increased oxidative stress and inflammation one week after 6-OHDA which resolved to baseline levels by twelve weeks, at which time pioglitazone-treated animals showed regionally preserved sympathetic innervation. Here we report post mortem characterization of heart and adrenal tissue in these animals compared to age and sex matched normal controls (n = 5). In the heart, 6-OHDA-treated animals showed a significant loss of sympathetic nerve fibers density (tyrosine hydroxylase (TH)-positive fibers). The anatomical distribution of markers of sympathetic innervation (TH) and inflammation (HLA-DR) significantly correlated with respective *in vivo* PET findings across left ventricle levels and regions. No changes were found in alpha-synuclein immunoreactivity. Additionally, CD36 protein expression was increased at the cardiomyocyte intercalated discs following PPARγ-activation compared to placebo and control groups. Systemic 6-OHDA decreased adrenal medulla expression of catecholamine producing enzymes (TH and aromatic L-amino acid

**Funding:** This study was supported by NIH P51OD011106 (nih.gov), NIH R21NS084158 [MEE] (nih.gov), NIH Kirschstein-NRSA F31HL136047 [JMM] (nih.gov), Welton L&S Honors Sophomore Scholarship [RAF] (honors.ls.wisc.edu), Trewartha Undergraduate Honors Research Grant [R.A.F] (honors.ls.wisc.edu), PF-APDA-SFW-1854 [HNM] (apdaparkinson.org), and the University of Wisconsin–Madison Office of Vice Chancellor for Research and Graduate Education (research.wisc.edu), Cellular and Molecular Pathology Graduate Program (cmp.wisc.edu), and Department of Medical Physics (medphysics.wisc.edu). The funders had no role in study design, data collection and analysis, decision to publish, or preparation of the manuscript.

**Competing interests:** The authors have declared that no competing interests exist.

decarboxylase) and circulating levels of norepinephrine, which were attenuated by PPARγ-activation. Overall, these results validate *in vivo* PET findings of cardiac sympathetic innervation, oxidative stress, and inflammation and illustrate cardiomyocyte CD36 upregulation as a marker of PPARγ target engagement.

## Introduction

Cardiac sympathetic neurodegeneration is common in Parkinson's disease (PD). Sixty percent of PD patients exhibit loss of postganglionic sympathetic innervation of the heart at the time of diagnosis [1], which progresses independent of motor symptoms [2, 3] to eventually affect 100% of the patient population [4]. PD cardiac sympathetic nerve loss has been documented *in vivo* as reduced cardiac [11C]meta-hydroxyephedrine (MHED) uptake [3, 5] and confirmed by post mortem evidence of loss of tyrosine hydroxylase (TH) immunostaining in cardiac nerve fibers and bundles [6–8]. Degeneration of the extrinsic sympathetic innervation to the heart produces fatigue and lowered cardiac contractility during exercise [9–11].

Approximately 30–40% of PD patients also suffer from orthostatic hypotension, the inability to regulate blood pressure with changes in body position, causing dizziness and syncope [12–14]. In addition to the loss of sympathetic innervation to the heart, PD patients with orthostatic hypotension show reduced circulating norepinephrine [15] and baroreflex failure [13]. The mechanisms leading to PD cardiac autonomic dysfunction are unclear and neuroprotective treatments are lacking. Accumulated evidence implicates oxidative stress and inflammation in PD neurodegeneration and suggests that these mechanisms could be targeted to induce neuroprotection [16, 17]. One potential neuroprotective strategy is activation of the peroxisome proliferator-activated receptor gamma (PPARγ) via pioglitazone administration [18]. Pioglitazone is a thiazolidinedione currently prescribed for its anti-diabetic, insulin-sensitizing action. In the last decade, pioglitazone has received much attention related to its ability to activate PPARγ and induce neuroprotection in a variety of neurological disease models [19, 20].

PD cardiac neurodegeneration can be modeled in rhesus macaques (*Macaca mulatta)* by intravenous administration of the catecholaminergic neurotoxin 6-hydroxydopamine (6-OHDA), which induces loss of cardiac sympathetic innervation and decreased circulating catecholamines [21, 22]. 6-OHDA selectively enters catecholaminergic neurons, autoxidizes, and interferes with mitochondrial complex I to produce reactive oxygen species. The oxidative damage leads to inflammatory cell recruitment that propagates neuronal loss [23, 24], similar to the role of inflammation and oxidative stress in PD neurodegeneration [16, 17, 25]. Our team recently reported the use of this model as a platform to develop methods for *in vivo* visualization of mechanisms of neurodegeneration and target validation of potential therapeutics for PD cardiac dysautonomia [26]. Positron emission tomography (PET) was performed with the radioligand MHED as a marker of sympathetic innervation, [61Cu]diacetyl-bis(N(4))-methylthiosemicarbazone (ATSM) for oxidative stress, and [11C]PBR28 (PBR28) for inflammation. Quantification of radioligand uptake over time identified significantly less oxidative stress and inflammation in the pioglitazone-treated animals compared to placebo at one week post-6-OHDA. Twelve weeks later, these values returned to baseline levels for both groups, while sympathetic innervation was regionally preserved in the pioglitazone-treated subjects.

Here, we report the post mortem characterization of heart and adrenal tissue in these animals compared to age and sex matched normal controls in order to validate the *in vivo* cardiac PET findings and further assess the effects of PPARγ activation.

## Materials and methods

### Ethics statement

The present study was performed in strict accordance with the recommendations in the National Research Council Guide for the Care and Use of Laboratory Animals (8[th] edition, 2011) in an AAALAC accredited facility (Wisconsin National Primate Research Center (WNPRC), University of Wisconsin-Madison). Experimental procedures were approved by the Institutional Animal Care and Use Committee (IACUC) of the University at the Wisconsin-Madison (experimental protocol G00705). The naïve (normal control) monkey tissues were obtained from the WNPRC tissue bank (original experimental protocols wprc00 and G005273). All efforts were made to minimize the number of animals used and to ameliorate any distress.

### Subjects

Cardiac and adrenal tissues from 15 adult, male rhesus monkeys (*Macaca mulatta*) were used in this project (S1 Table). The animals were individually housed in Group 3 or Group 4 enclosures (cage floor area 4.3 ft.$^2$ or 6.0 ft.$^2$ per animal, height 30 or 32 in.) in accordance with the Animal Welfare Act and its regulations and the Guide for the Care and Use of Laboratory Animals (8th edition, 2011) with a 12-hour light/dark cycle and room temperature of 21˚C. Throughout the study, the animals were monitored twice daily by an animal research or veterinary technician for evidence of disease or injury (e.g., inappetence, dehydration, diarrhea, lethargy, trauma, etc.), and body weight was monitored to ensure animals remained in properly sized cages. Animals were fed commercial nonhuman primate chow (2050 Teklad Global 20% Protein Primate Diet, Harlan Laboratories, Madison, WI) twice daily, supplemented with fruits or vegetables and a variety of forage items and received ad libitum water. Nonhuman primate chow soaked in a protein-enriched drink (Ensure©, Abbott Laboratories, Abbott Park, IL) was offered to stimulate appetite as needed. Environmental enrichment opportunities were presented a minimum of 5 days per week. Enrichment toys and puzzles were provided on a rotational schedule to ensure novel stimulation and to engage animals' species-typical curiosity and manipulatory behavior.

Ten animals were part of a previously published study [26]. They received systemic 6-OHDA (50mg/kg i.v.), as previously described [22]. For the procedure, the monkeys were food-deprived overnight; anesthesia was induced with ketamine HCl (15mg/kg i.m.) and maintained with 1–3% isoflurane in 100% O$_2$ at 1 L/min under constant veterinarian monitoring. Twenty-four hours later, the animals were randomly and blindly assigned to receive daily oral dosing of placebo (n = 5; mean age = 8.5 years old; age range = 6.2–13 years old; mean weight = 10.8kg; weight range = 9.8–12.3 kg) or pioglitazone (5mg/kg; n = 5; mean age = 7.0; age range = 5.6–11.4 years old; mean weight = 10.1kg; weight range = 9.4–10.6 kg). Pioglitazone was administered in the home cage in the morning at the time of AM feeding and was given orally by crushing the pill and placing it in a high value food item; placebo treatment consisted of the same high value food item. Cardiac PET imaging with MHED, PBR28, and ATSM was performed at baseline, one week, and twelve weeks post-neurotoxin administration. Twenty-four hours after the last PET was acquired, the animals were euthanized.

The normal control rhesus monkey tissues (n = 5; mean age = 8.8 years old; age range = 6.8–12.4 years old; mean weight = 10.1kg; weight range = 9.6–12.6 kg) were obtained from the WNPRC tissue bank. Donors were selected by matching them to the baseline condition of the 6-OHDA-treated monkeys (normal adult rhesus monkeys without previous history of cardiac dysfunction). The number of subjects per group was defined by our previous cardiac study [21, 22] in which n = 5 found *in vivo* a statistically significant loss of cardiac denervation.

## Necropsy and tissue preparation

All monkeys (6-OHDA-treated and normal controls) were euthanized by transcardiac perfusion and tissues collected following previously published and approved methods [21, 22, 26]. Briefly, all animals were anesthetized with ketamine hydrochloride (10mg/kg, i.m.) followed by pentobarbital sodium (minimum of 25mg/kg, i.v.) and perfused through the left atrium with heparinized phosphate-buffered saline, followed by 4% paraformaldehyde. Hearts and adrenals were post-fixated in 4% paraformaldehyde for 24 hours followed by 70% ethanol. For paraffin embedding, hearts were cut with the aid of a calibrated polymethyl methacrylate slice apparatus. The hearts were laid on the inferior cardiac wall and sliced into 4mm sections in a transverse plane creating 8 levels from the base to the apex (S1 Fig), matching the orientation of the left ventricle in the published PET imaging study [26]. A 1mm punch was placed in the anteroseptal myocardium to mark orientation and sections were blocked in paraffin. Three cardiac sections were analyzed in this study, representing the base, middle, and apex cardiac levels (S1 Fig). Adrenals were cut through the middle of the gland and blocked in paraffin with the cut-face up. All blocked tissue was cut on a standard rotary microtome in 5μm section thickness and mounted on positively charged slides.

## General immunohistochemistry

Cardiac immunohistochemistry against tyrosine hydroxylase (TH), alpha-synuclein (α-syn), 8-hydroxy-2'-deoxyguanosine (8-OHdG), human leukocyte antigen DR (HLA-DR), peroxisome proliferator-activated receptor gamma (PPARγ) coactivator 1-alpha (PGC1α), and cluster of differentiation 36 (CD36) was performed following previously published methods [21, 26] (S2 Table). Briefly, cardiac sections were deparaffinized and treated for heat antigen retrieval in a microwave for 6 minutes at 100% power followed by 6 minutes at 80% power and left to cool for 1hr at room temperature. The sections were then washed and endogenous peroxidase activity blocked by incubation of 30% $H_2O_2$ and methanol. Nonspecific binding sites were blocked with either Super Block (ScyTek, Logan, UT) or appropriate species serum for 30 minutes at room temperature and incubated overnight with a primary antibody diluted in blocking buffer plus 0.1% Triton-X (S2 Table). The sections were then incubated in appropriate biotinylated secondary antibody (1:200), followed by avidin-biotin-complex peroxidase (VECTASTAIN Elite ABC HRP Kit, Vector Laboratories, Burlingame, CA; all antibodies other than 8-OHdG) or avidin-biotin-complex alkaline phosphatase kit (VECTASTAIN ABC-AP Staining Kit, Vector Laboratories, Burlingame, CA; 8-OHdG), and visualized with either a commercial 3,3'-diaminobenzidine (DAB) kit (Vector Laboratories, Burlingame, CA; all antibodies other than 8-OHdG) or VectorBLACK kit (8-OHdG) (Vector Laboratories, Burlingame, CA). Cardiac sections were counterstained with either Hematoxylin Stain (VWR, Radnor, PA) or Nuclear Fast Red (Vector Laboratories, Burlingame, CA) or left without counterstain, depending on the immunostaining (S2 Table), dehydrated, and coverslipped (Cytoseal mounting media, Thermo Scientific, Waltham, MA).

Adrenal sections were deparaffinized and treated for heat antigen retrieval in a microwave for 6 minutes at 100% power followed by 6 minutes at 80% power and left to cool for 30 minutes at room temperature. Adrenal tissue sections were immunostained against TH and aromatic L-amino acid decarboxylase (AADC) following the procedure described above (S2 Table).

Tissues from all three groups were processed in parallel for each staining to minimize bias. Negative controls for cardiac and adrenal immunostainings were performed in parallel by omitting the primary antibodies in the immunostaining procedures. Cardiac and adrenal tissues were also stained for hematoxylin and eosin (HE) for general anatomical evaluations.

## Immunofluorescence

Immunofluorescence stainings were performed to identify cell types in cardiac nerve bundles expressing HLA-DR, including non-myelinating Schwann cells (colocalization of HLA-DR and S100 calcium-binding protein B (S100B)) and macrophages (colocalization of HLA-DR and CD68) (S3 Table). Slides were deparaffinized and antigen retrieval performed as described above. Tissue was blocked with 5% donkey serum and 2% BSA solution and incubated overnight at 4°C in primary antibodies (S3 Table) diluted in blocking buffer plus 0.1% Triton-X. The sections were then incubated with Alexafluor-conjugated secondary antibody (1:1000) against the appropriate species and coverslipped with mounting media with DAPI (Vector Laboratories, Burlingame, CA).

## Anatomical evaluation

Board-certified WNPRC veterinary pathologists specializing in nonhuman primate models and diseases (HAS, AM) blindly evaluated three HE stained cardiac sections, representing the base, middle, and apex cardiac levels (S1 Fig), and at least 1 section of adrenal gland per animal. Histological findings, such as the presence and severity of inflammation, atrophy, or mineral deposits, were recorded and reported.

## Quantification of cardiac immunostaining

For detailed anatomical quantification of immunohistochemical markers, the heart was divided into 3 cardiac levels (base, middle, and apex) and each level then subdivided into four regions (septal, anterior, lateral, inferior; S1 Fig).

**TH expression.** Sympathetic innervation in the left ventricular myocardium was identified as TH-immunoreactivity (-ir) in nerve bundles (collections of axons) and individual nerve fibers as previously described [21]. Quantification of TH-ir in the cardiac nerve bundles of these subjects was previously reported [26]. Briefly, percent area above threshold (%AAT) and optical density (OD) of TH-ir were calculated using NIH ImageJ software in six nerve bundles per region at the three cardiac levels described above (S1 Fig). Photomicrographs of myocardial nerve bundles with areas >30μm$^2$ were captured at 63x with a Zeiss Axioimager M2 equipped with a Qimaging camera. DAB color was separated from hematoxylin counterstain with ImageJ Colour Deconvolution filter. ImageJ was calibrated using a step tablet, grey scale values were converted to OD units using the Rodbard function, bundles were outlined to define the region of interest (ROI), and mean OD and %AAT measured using a threshold 0.68. Mean OD is the mean grey value, or darkness, of the immunoreactivity of all pixels in the ROI, and %AAT is the percentage of the pixels in the ROI that have a mean grey value above a predetermined threshold.

TH-ir nerve fiber density was quantified in each cardiac region and level described above (S1 Fig) using a Zeiss Axioimager M2 microscope and the area fraction fractionator (AFF) probe function in StereoInvestigator v10.0 (MicroBrightField, Williston, VA) as previously described [21]. ROIs were outlined under low magnification (2.5x) for each cardiac region of the left ventricle. ROIs were defined as 8mm wide and traced by drawing perpendicular lines from the left ventricular lumen to the epicardium joined by lines following the epicardial and endocardial borders (mean ROI = 38.8mm$^2$). Using a 63x oil immersion objective, the AFF randomly placed the 0.0135mm$^2$ counting frame within the ROI and continued throughout the tissue moving with x and y sampling grids of 700μm. In each counting frame, a 2μm grid was placed, and an investigator highlighted the area of the grid with immunostained nerve fibers. TH-ir in nerve bundles >30μm$^2$ was excluded during fiber area density procedure. TH-

ir fiber density was calculated as the percentage of the total area of ROI ($\mu m^2$) with TH-ir fibers ($\mu m^2$) [(TH-ir fiber area ($\mu m^2$) / total area of ROI ($\mu m^2$)) * 100].

**α-Syn expression.**　To assess for the presence of PD-like α-syn accumulation or aggregation, %AAT and OD of α-syn-ir were calculated in six nerve bundles per region at the three cardiac levels (S1 Fig) using Image J software as described for TH-ir quantification in cardiac bundles.

**8-OHdG expression.**　8-OHdG-ir was used as a marker of oxidative stress. It was evaluated in myocardial nerve bundles and cardiomyocytes in each region in the base cardiac level (S1 Fig). Analysis was limited to this level to better match available PET data with ATSM, which was affected by the high uptake of this radioligand by the liver (Metzger et al 2018). %AAT and OD of 8-OHdG-ir in myocardial nerve bundles were calculated using Image J software as described above, using a threshold of 0.18. VectorBLACK color was separated from Fast Red counterstain with ImageJ Colour Deconvolution filter. Three bundles were analyzed per region.

A semiquantitative rating scale (S2 Fig) was used to quantify 8-OHdG-ir in cardiomyocyte nuclei. The cardiac sections were evaluated at 10x magnification and assigned a score as follows: 0 = no to very little stain (<10% nuclei have any stain); 1 = diffuse light stain 10–100% of nuclei OR 10–20% nuclei medium/dark stain; 2 = >20% nuclei medium/dark stain; 3 = >50% nuclei medium/dark stain. Ratings were performed by two individuals (Pearson correlation to test inter-rater reliability: r = 0.9; p < 0.0001), which were averaged together for data analysis.

**HLA-DR expression.**　HLA-DR-ir was used to identify antigen presenting cells. %AAT and OD of HLA-DR-ir in myocardial nerve bundles were calculated using Image J software as described above, using a threshold of 0.44. Three bundles were analyzed per region at the base cardiac level.

Semiquantitative rating scales were also used to evaluate HLA-DR expression in three different parts of the cardiac tissue (S3 Fig): in microvascular endothelial cells (capillary HLA-DR), adjacent to blood vessels (perivascular HLA-DR; vessels >80 $\mu m$ diameter), as well as in nerve bundles (nerve bundle HLA-DR). These ratings were performed in each cardiac region and level (S1 Fig). Capillary HLA-DR-ir was rated from 0 to 3 based on the percent of the area of each region (septal, anterior, lateral, inferior) that exhibited HLA-DR positive capillaries as follows: 0 = <50% area of the region has HLA-DR-ir capillaries; 1 = >50% area of the region has HLA-DR-ir capillaries; 2 = 100% of the area of the region has light HLA-DR-ir capillaries; 3 = 100% of the area of the region has medium/dark HLA-DR-ir capillaries. Perivascular HLA-DR-ir was rated from 0 to 3 based on the number of HLA-DR-ir cells counted within 350$\mu m$ of the vessel as follow: 0 = 0 HLA-DR-ir cells; 1 = 1 to 5 HLA-DR-ir cells; 2 = 6 to 10 HLA-DR-ir cells; 3 = >10 HLA-DR-ir cells. Nerve bundle HLA-DR-ir was rated in every nerve bundle >30 $\mu m^2$ (mean of 14 bundles in each region) from 0 to 3 based on the number of HLA-DR-ir cells counted in each bundle as follows: 0 = 0 HLA-DR-ir cells in the bundle; 1 = 1 or 2 HLA-DR-ir cells in the bundle; 2 = 3 to 5 HLA-DR-ir cells in the bundle; 3 = >5 HLA-DR-ir cells in the bundle.

**PPARγ target engagement.**　Pioglitazone target engagement was assessed by immunohistochemistry for PGC1α and CD36. PGC1α-ir was quantified in nerve bundles and cardiomyocytes in each region in the middle cardiac level (S1 Fig). For analysis in nerve bundles, %AAT and OD of PGC1α-ir were calculated in three bundles per region using Image J software as described above, using a threshold of 0.37. For analysis in cardiomyocytes, three photomicrographs of cardiomyocytes were captured per region at 40x with a Zeiss Axioimager M2 equipped with a Qimaging camera. DAB color was separated from hematoxylin using ImageJ as described above, ROIs were drawn around cardiomyocytes to exclude areas of blood vessels and tissue separation, and OD and %AAT were measured using a threshold of 0.15.

CD36-ir was assessed in cardiomyocytes in each region in the middle cardiac level (S1 Fig) using both ImageJ Software and a semiquantitative rating scale (S4 Fig). For both analyses, five

cardiomyocyte images per region were captured at 40x as described above; to ensure the intercalated discs were visible, image capture was limited to longitudinally-oriented cardiomyocytes (e.g. transverse-oriented cardiomyocytes were not analyzed).

OD and %AAT of CD36-ir in cardiomyocytes were calculated with ImageJ software using a threshold of 0.28.

Ratings of CD36-ir were focused on the level of expression at the intercalated discs, the part of the sarcolemma, or cardiomyocyte cell membrane, where two cardiomyocytes connect (S4 Fig). CD36-ir was rated in each image as follows (S4 Fig): 0 = no to very little CD36-ir; 1 = CD36-ir present very lightly in 10–100% of the intercalated disks AND/OR a few (2–4) discs have medium/dark CD36-ir; 2 = <1/3 of the area shows regularly visible CD36-ir at the intercalated discs; 3 = >1/3 of the area shows regularly visible CD36-ir at the intercalated discs. To validate the reliability of the rating scale, intra-rater relatability was calculated by having the same investigator rate each image twice while blind to the previous rating (Pearson correlation to test intra-rater reliability: r = 0.93; p < 0.0001); an average of the two ratings were used for analysis.

## Quantification of adrenal immunostaining

Expressions of TH and AADC were each quantified blindly in three adrenal slides per animal. Images were captured at low magnification (2.5x) using an Axioimager M2 equipped with a Qimaging camera as previously described [21]. Using ImageJ software, 3 ROIs were drawn per slide throughout the adrenal medulla, avoiding any tissue processing artifacts (e.g. tears, folds, etc.) and vasculature >2.5 $\mu m^2$. OD and %AAT quantifications were collected using methods similar to the cardiac evaluations, with a threshold of 0.2.

## High performance liquid chromatography (HPLC)

Plasma norepinephrine, dopamine, epinephrine, and dihydroxyphenylacetic acid (DOPAC) were assayed using high-performance liquid chromatography (HPLC) with electrochemical detection (ESA, Chelmsford, MA) following previously described methods [22] with modifications. Blood samples for HPLC were obtained at baseline, 1, and 12 weeks after toxin administration. Blood was collected in a K2 EDTA tube, immediately mixed with 10% sodium metabisulfite (0.7%), and centrifuged. 1.0mL of plasma was extracted using alumina (aluminum oxide) and analyzed using a coulometric electrochemical detector (Choulochem III; ESA, Chelmsford, MA). Every 1L of mobile phase used to separate the catecholamines contained 13.8g of sodium phosphate, 55mg of 1-octane sulfonic acid, 55mg of EDTA, and 45mL of acetonitrile at pH of 3.85. Mobile phase was filtered through a 0.22μm GV filter under vacuum and pumped into the system at a rate of 1.0mL/min, producing a pressure of approximately 180 bars. The electrodes were set at −250 and 380mV. Peak areas were measured from the chromatograms and compared to the peak areas for extracted calibrators. Calibrators ranged from 0.9ng/ml to 500ng/ml and linearity was >0.97. The lower limit of quantification for all analytes was set at 3.9ng/ml; no samples were found to be below the detectable HPLC sensitivity. A small number of samples were not included in the analysis due to technical difficulties (S6R Table). Coefficient of variation was 3.4% for norepinephrine, 6.4% for epinephrine, 16.1% for dopamine, and 17.2% for DOPAC.

## Statistical analysis

Data collection and analysis were performed by investigators blind to the treatment groups. Statistical analysis was performed using GraphPad Prism (version 8.0, GraphPad Software), R

(3.2.4), or SPSS (version 24). A p value <0.05 was accepted as significant. A p value of <0.1 was accepted as a trend. Averages are mean ± SEM unless otherwise specified.

For cardiac data analysis, comparisons between treatment groups, cardiac level, and cardiac region were analyzed using ANOVA as 3 (Treatment: 6-OHDA + pioglitazone vs. 6-OHDA + placebo vs. normal control) × 3 (Level: base, middle, apex) × 4 (Regions: septal, anterior, lateral, inferior) with repeated measures on Levels and Regions, post hoc analysis using Bonferroni multiple comparisons, and Huynh–Feldt adjusted p values for repeated measures to account for potential violations of the sphericity assumption. For data collected at only one level, comparisons were carried out using similar methods but with ANOVA as 3 (Treatment: 6-OHDA + pioglitazone vs. 6-OHDA + placebo vs. normal control) × 4 (Regions: septal, anterior, lateral, inferior). ANOVA main effect sizes are presented as partial eta squared ($\eta_p^2$) calculated in SPSS. Data were visually examined for outliers and reasonable normality using the qq plot feature of R software.

Correlations between 1) TH AFF fiber density and 12 week MHED PET or TH-ir in nerve bundles, 2) HLA-DR-ir nerve bundle rating and %AAT or OD, 3) HLA-DR ratings and 12 week PBR28 PET, and 4) CD36-ir intercalated disc rating and CD36-ir cardiomyocyte %AAT or OD were performed following previously described methods of repeated measures correlations ($r_{rm}$) [27] in R (v3.4.2). Briefly, repeated measures correlations are appropriate when there are multiple observations on each subject such that use of the standard Pearson correlation is inappropriate. The 'rmcorr' package in R fits a linear set of relations between quantitative variables that allow the intercepts to vary across animals while holding the slope constant. For repeated measures correlations, bootstrap confidence intervals were calculated in addition to the parametric confidence intervals. 12 week MHED and PBR28 PET data included in the repeated measures correlations were matched to levels and regions analyzed by post mortem immunohistochemistry as previously described (PET levels 1, 4, and 6; PET regions SA, AL, LI, and IS) [26]. Repeated measures correlation between capillary HLA-DR-ir rating and 12 week PBR28 PET was not performed because the rating given per region generated a data set with discrete values of 0, 1, 2, or 3 for each region, which is not suitable for correlation with a continuous variable. Correlations between 12 week ATSM PET data and 8-OHdG-ir were performed using Pearson's correlation because only one ROI was analyzed by ATSM PET due to parts of the heart being obscured by radioligand uptake in the liver as previously described [26]; 8-OHdG-ir data used for these correlations were an average of the four regions analyzed for each animal. One animal in the 6-OHDA + pioglitazone group was excluded from 12-week ATSM data due to subcutaneous radioligand administration [26]. In case of possible violations of normality in datasets evaluated by Pearson's correlations, Spearman's rank order correlations were also performed and are reported.

For adrenal medulla data analysis, comparisons between groups used 1 way ANOVA followed by post hoc analysis with Bonferroni multiple comparisons. One way ANOVA effect sizes are presented as eta squared ($\eta^2$) calculated in SPSS.

HPLC data were analyzed in R by comparing between treatment groups (Mann-Whitney) and between time points (Wilcoxon signed-rank). For comparing between treatment groups, data were normalized by subtracting the baseline value from the 1 or 12-weeks post-toxin values, dividing by the baseline value, and then multiplying by 100; these % change from baseline values used for analysis. Effect sizes are reported as r = Z/(sqrt(N)).

## Results

### Anatomical evaluation

Evaluation of HE stained tissue sections showed typical cardiac (S4 Table) and adrenal (S5 Table) morphology.

In cardiac tissue, cardiomyocytes presented with centrally located nuclei and visible cross-striations and intercalated disks. Cardiac nerve bundles containing nerve fibers surrounded by epineurium were found throughout the myocardium (S5 Fig). Minimal to mild lymphocytic infiltration surrounding vasculature and in the myocardium was variably present (S4 Table). In the 6-OHDA + placebo group, one animal presented minimal lymphocytic myocarditis (Placebo 1) and one animal exhibited minimal focal neutrophilic myocarditis (Placebo 4). In the 6-OHDA + pioglitazone group, one animal showed minimal multifocal lymphocytic myocarditis, mild to moderate degenerative and fibrotic cardiomyopathy, and mild multifocal adipocyte infiltration (Pioglitazone 4). These findings are not uncommon in rhesus macaques of this age and are typically not associated with clinical cardiac dysfunction [28].

The adrenal medulla displayed high vascularity, with the capillaries surrounded by columnar chromaffin cells exhibiting a typical granular cytoplasm (S5 Fig). No significant lesions were found except in one 6-OHDA + pioglitazone animal which showed lymphocytic infiltration, medullary atrophy, vacuolated medullary cells, and loss of medullary cells (Pioglitazone 1; S5 Table). Follow up evaluation of Prussian blue and TH stained tissue confirmed medullary cell loss indicated by absence of TH-ir in medullary cells; this was not associated with iron deposition or fibrosis.

## TH expression

The sympathetic marker TH was present in cardiac nerve fibers grouped together in nerve bundles and as individual fibers running through the myocardium. In nerve bundles, %AAT and OD of TH-ir were significantly decreased in every cardiac region and level in both the 6-OHDA + placebo and 6-OHDA + pioglitazone groups compared to normal controls, as previously reported [26]; no statistically significant differences were observed between the placebo and pioglitazone groups. TH-ir nerve fibers in the cardiac left ventricular myocardium exhibited typical pearl necklace-like morphology in longitudinal orientation or single punctate structures in transverse orientation (Fig 1). TH-ir fiber density was significantly decreased by 6-OHDA three months post-toxin (overall treatment effect, $p < 0.00005$, $\eta_p^2 = 0.813$; treatment x level effect, $p < 0.05$, $\eta_p^2 = 0.340$). The 6-OHDA-induced decrease in TH-ir fiber density was significant in both toxin-treated groups compared to control when averaging over the entire heart (control vs. 6-OHDA + placebo, $p < 0.0002$; control vs. 6-OHDA + pioglitazone, $p < 0.0002$) and also at each individual cardiac level [(control vs. 6-OHDA + placebo: Base, $p < 0.02$; Mid, $p < 0.008$; Apex, $p < 0.00009$) and (control vs. 6-OHDA + pioglitazone: Base, $p < 0.006$; Mid, $p < 0.02$; Apex, $p < 0.0002$)] and region [(control vs. 6-OHDA + placebo: S, $p < 0.0003$; A, $p < 0.0006$; L, $p < 0.002$; I, $p < 0.00007$) and (control vs. 6-OHDA + pioglitazone: S, $p < 0.0004$; A, $p < 0.0006$; L, $p < 0.002$; I, $p < 0.0003$)] (Fig 1). No significant differences were found between the placebo and pioglitazone groups. Within individual treatment groups, only the control group showed an effect of left ventricle anatomy on sympathetic fiber density; TH-ir fiber density was significantly higher in the septal region than the inferior region ($p < 0.05$) (Fig 1).

Repeated measures correlations were performed to assess the relationship between post mortem (TH-ir) and *in vivo* (12 week PET with MHED radioligand) measures of sympathetic innervation. TH-ir %AAT in cardiac nerve bundles and 12 week MHED uptake significantly correlated across cardiac levels and regions ($r_{rm} = 0.44$, $p < 0.000002$), as previously reported [26]. Similarly, the anatomical distribution of sympathetic fiber density also correlated across cardiac regions and levels with 12 week MHED uptake ($r_{rm} = 0.516$, $p < 0.000000007$), in addition to correlating with nerve bundle TH-ir %AAT ($r_{rm} = 0.399$, $p < 0.0000001$) and OD ($r_{rm} = 0.384$, $p < 0.0000003$) (Fig 1). All TH-ir fiber density correlations remained significant

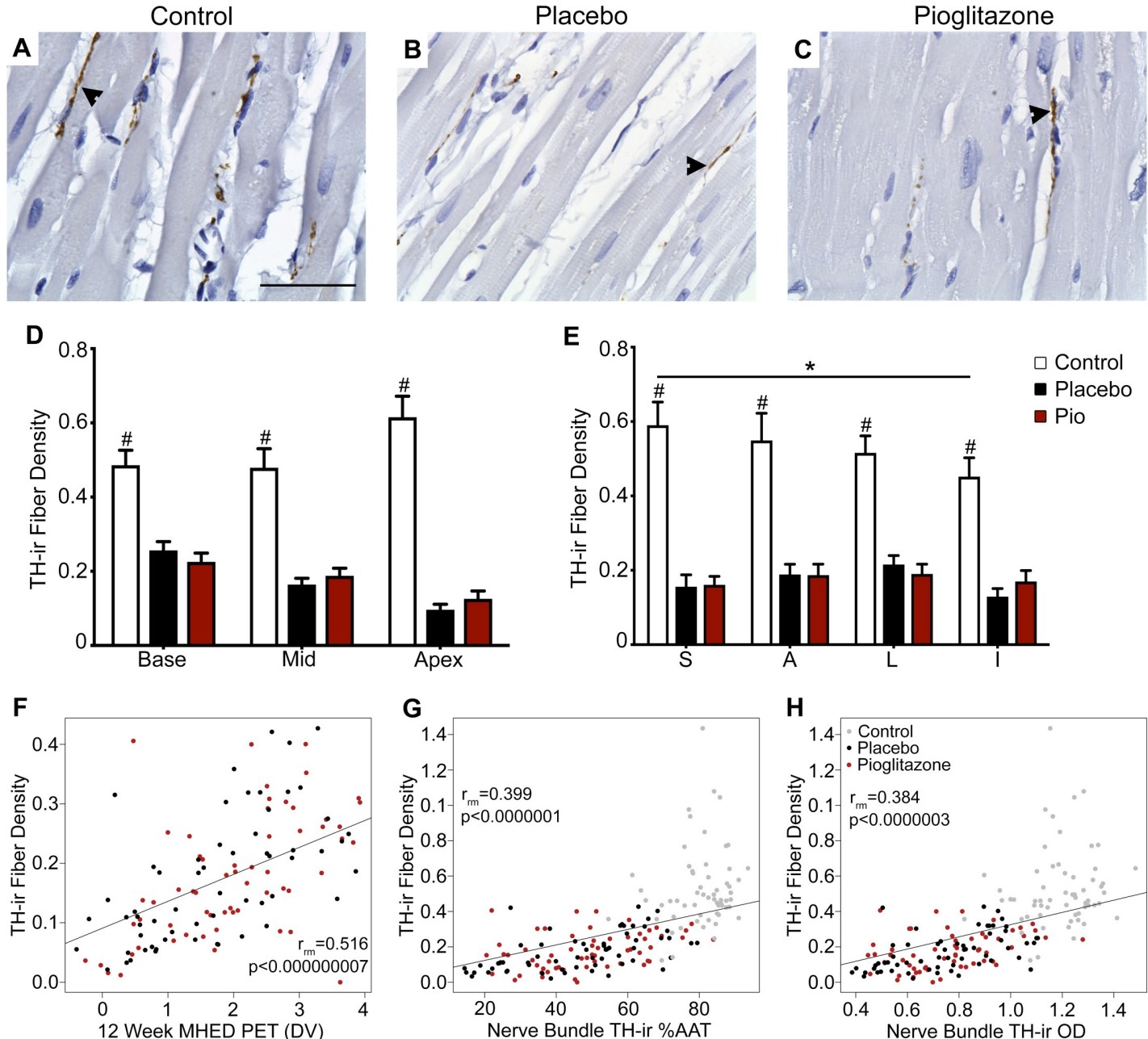

**Fig 1. Cardiac sympathetic fiber density decreased post-6-OHDA and correlated with *in vivo* and post mortem markers.** (A-C) Photomicrographs of TH-positive myocardial sympathetic nerve fibers in the (A) control, (B) 6-OHDA + placebo, and (C) 6-OHDA + pioglitazone groups. Scale bar = 50 μm. Arrowheads point to TH-positive nerve fibers. (D) TH-ir fiber density at three cardiac levels (each level is an average of the four regions in that level). (E) TH-ir fiber density at four cardiac regions (each region is an average of the three levels in that region). (F-H) Plots of repeated measures correlations across cardiac regions and levels of left ventricle TH-ir fiber density with (F) *in vivo* evaluation of sympathetic innervation at 12 week post-6-OHDA as measured by MHED PET and nerve bundle TH-ir (G) %AAT and (H) OD. Each point represents a single anatomical area in one animal (12 total anatomical areas (3 levels * 4 regions) per animal; (F) 10 animals or (G, H) 15 animals). Error bars = SEM. #, control group significantly different from the 6-OHDA + placebo and 6-OHDA + pioglitazone groups. *, significant difference between two anatomical areas within one treatment group; see text for details. 6-OHDA, 6-hydroxydopamine; TH-ir, tyrosine hydroxylase immunoreactivity; TH-ir fiber density, percentage of myocardial area with TH-ir nerve fibers; Pio, pioglitazone; Mid, middle; S, septal; A, anterior; L, lateral; I, inferior; %AAT, percent area above threshold; OD, optical density; MHED, [11C]meta-hydroxyephedrine; PET, positron emission tomography; DV, distribution volume.

when each treatment group was analyzed individually, and bootstrap confidence intervals were very similar to parametric confidence intervals (S11 Fig; S7 Table).

### α-Syn expression

α-Syn-ir was present in nerve bundles and fibers of the cardiac left ventricle (S6 Fig). Throughout the nerve bundles, α-syn-ir was visible as a diffuse shading and as darker punctate spots. Qualitatively, α-syn-ir appeared darker in the normal, healthy control animals than those treated with 6-OHDA. However, quantification of %AAT and OD of α-syn-ir in cardiac nerve bundles did not show statistically significant effects of treatment (%AAT treatment effect, $p > 0.2$, $\eta_p^2 = 0.233$; OD treatment effect, $p > 0.2$, $\eta_p^2 = 0.228$).

### 8-OHdG expression

Expression of the oxidative stress marker 8-OHdG was variably present as dark grey to black shading within nuclei in cardiac nerve bundles and cardiomyocytes (Fig 2). In cardiac nerve bundles, %AAT of 8-OHdG-ir showed a trend ($p < 0.07$, $\eta_p^2 = 0.367$) toward an overall effect of treatment, which was significant when treatment was considered in combination with region (treatment x region: %AAT, $p < 0.04$, $\eta_p^2 = 0.309$; OD, $p < 0.03$, $\eta_p^2 = 0.312$). Post hoc analysis revealed that only the inferior region of the heart had elevated oxidative stress in both 6-OHDA-treated groups compared to the control group [(control vs. 6-OHDA + placebo: %AAT, $p < 0.002$; OD, $p < 0.02$) and (control vs. 6-OHDA + pioglitazone: %AAT, $p < 0.00005$; OD, $p < 0.003$)] (Fig 2). Ratings of 8-OHdG-ir in cardiomyocytes were not significantly different between groups (treatment effect, $p > 0.30$, $\eta_p^2 = 0.176$) (Fig 2).

Previous PET *in vivo* evaluation of oxidative stress in these animals found increased cardiac ATSM levels at one week post-neurotoxin, which returned to baseline values at 12 weeks [26]. No significant correlations were found between 12 week ATSM data and 8-OHdG expression in cardiac nerve bundles [(Pearson- %AAT: $r = -0.4$, $p > 0.29$; OD: $r = -0.21$, $p > 0.59$) and (Spearman- %AAT: $r = -0.37$, $p > 0.34$; OD: $r = -0.18$, $p > 0.64$)] or in cardiomyocytes [(Pearson: $r = -0.18$, $p > 0.64$) and (Spearman: $r = -0.29$, $p > 0.46$)] (Fig 2).

### HLA-DR expression

The inflammatory marker HLA-DR was expressed in nerve bundles (nerve bundle HLA-DR-ir), microvascular endothelial cells (capillary HLA-DR-ir), and cells adjacent to blood vessels (perivascular HLA-DR-ir) (Fig 3). Consistent with analysis of HE-stained sections, inflammation was minimal and at normal background levels, with the exception of one animal in the 6-OHDA + pioglitazone group (S4 Table).

Analysis of HLA-DR-ir in nerve bundles of the base of the heart (S7 Fig) showed a trend toward an overall effect of treatment group for %AAT ($p < 0.07$, $\eta_p^2 = 0.376$), but not for OD ($p > 0.9$, $\eta_p^2 = 0.007$). Post hoc analysis indicated no significant differences between any individual treatment groups [(control vs. 6-OHDA + placebo: %AAT, $p = 1$; OD, $p = 1$) and (6-OHDA + placebo vs. 6-OHDA + pioglitazone: %AAT, $p > 0.136$; OD, $p = 1$)], although a trend was observed between control and 6-OHDA + pioglitazone groups only for %AAT (% AAT, $p < 0.097$; OD, $p = 1$).

To further investigate subtle differences between treatment groups, HLA-DR expression was scored in all cardiac nerve bundles across the 3 levels of the heart with a semiquantitative rating scale designed to discriminate between zero and mild HLA-DR-ir (Fig 3; S3 Fig). The ratings showed that HLA-DR-ir in nerve bundles was significantly affected by treatment ($p < 0.05$, $\eta_p^2 = 0.396$), treatment in combination with region ($p < 0.03$, $\eta_p^2 = 0.327$), region combined with level ($p < 0.006$, $\eta_p^2 = 0.223$), and treatment combined with region and level

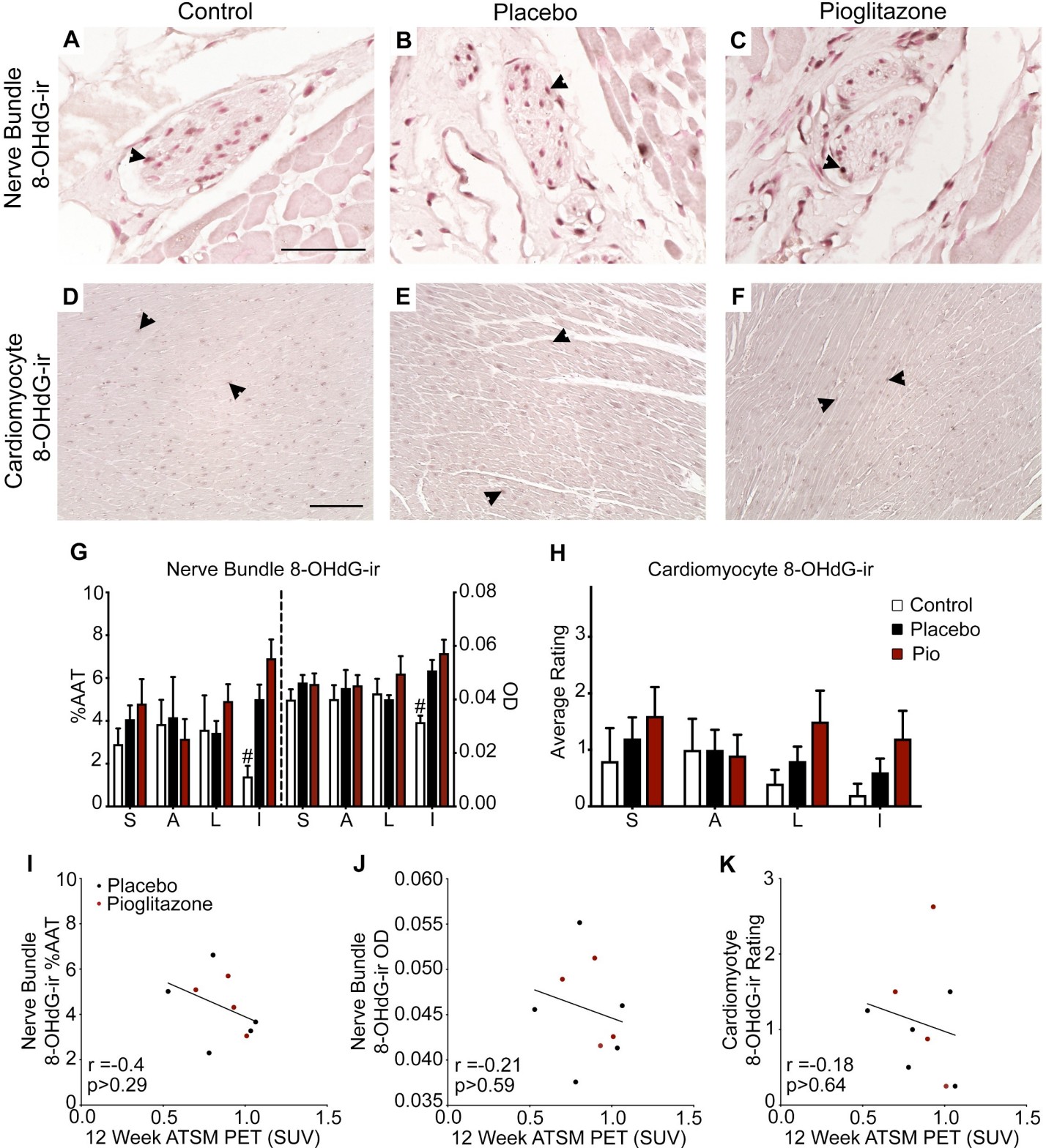

**Fig 2. Oxidative stress was minimally increased 12 weeks post-6-OHDA and did not correlate with *in vivo* findings.** (A-F) Photomicrographs of left ventricle (A-C) nerve bundles or (D-F) cardiomyocytes immunostained for the nuclear oxidative stress marker 8-OHdG in the (A, D) control, (B, E) 6-OHDA + placebo, and (C, F) 6-OHDA + pioglitazone groups. Scale bar = (A) 50 μm and (D) 200 μm. Arrowheads point to 8-OHdG-ir nuclei. (G) 8-OHdG-ir in cardiac nerve bundles. (H) 8-OHdG-ir

in cardiomyocytes. (I-K) Plots of Pearson's correlation between 12 week cardiac ATSM uptake and left ventricle nerve bundle 8-OHdG (I) %AAT and (J) OD and (K) cardiomyocyte 8-OHdG rating. Error bars = SEM. #, control group significantly different from the 6-OHDA + placebo and 6-OHDA + pioglitazone groups; see text for details. 6-OHDA, 6-hydroxydopamine; 8-OHdG-ir, 8-hydroxy-2'-deoxyguanosine immunoreactivity; Pio, pioglitazone; %AAT, percent area above threshold; OD, optical density; S, septal; A, anterior; L, lateral; I, inferior; PET, positron emission tomography; SUV, standard uptake volume; ATSM, [61Cu]diacetyl-bis(N(4))-methylthiosemicarbazone.

($p < 0.006$, $\eta_p^2 = 0.311$). Comparing groups, there was a trend toward a significant difference between the control and the 6-OHDA + pioglitazone group ($p < 0.058$), but no difference between the control and 6-OHDA + placebo group ($p = 1.0$) or the 6-OHDA + pioglitazone and 6-OHDA + placebo group ($p > 0.2$). Detailed analysis of effects involving condition showed that the 6-OHDA + pioglitazone group had higher HLA-DR-ir nerve bundle rating than the control group at the lateral ($p < 0.03$) and inferior ($p < 0.04$) regions and showed a trend toward higher rating than the 6-OHDA + placebo group at the lateral region ($p < 0.07$; Fig 3). Within the 6-OHDA + pioglitazone group, HLA-DR-ir rating was significantly higher in nerve bundles in the lateral than the anterior region of the myocardium ($p < 0.0008$; Fig 3).

Next, fluorescent co-labeling was performed to investigate which cell types were expressing HLA-DR in nerve bundles. CD68-ir (a macrophage marker) was present in cells adjacent to vessels in cardiac tissue but was not observed in nerve bundles in any animals (S8 Fig). S100B (a marker of myelinating and non-myelinating Schwann cells) was occasionally observed co-labeling with HLA-DR in a few (1–3) nerve bundles in all animals (S9 Fig).

In microvascular endothelial cells of capillaries, ratings of HLA-DR-ir were not different between treatments when averaged over all regions and levels ($p > 0.61$, $\eta_p^2 = 0.08$) but were significantly affected by cardiac region ($p < 0.03$, $\eta_p^2 = 0.237$) and combined region and level ($p < 0.04$, $\eta_p^2 = 0.170$). No significant differences were found between any individual regions or conditions in any treatment group (Fig 3).

Perivascular HLA-DR-ir ratings were also not affected by treatment ($p > 0.492$, $\eta_p^2 = 0.112$), and there were no significant interactions involving treatment. However, the distribution of perivascular HLA-DR-ir cells was substantially dependent on cardiac anatomy as illustrated by significant effects of region ($p < 0.006$, $\eta_p^2 = 0.29$), level ($p < 0.00016$, $\eta_p^2 = 0.515$), and combined region and level ($p < 0.04$, $\eta_p^2 = 0.175$) when analyzed with all treatment groups combined. This difference was most apparent between levels, with significantly higher perivascular HLA-DR-ir in the base of the heart compared to both the middle ($p < 0.01$) and the apex ($p < 0.015$) in the 6-OHDA + pioglitazone group and a trend toward higher HLA-DR in the base compared to the apex in the 6-OHDA + placebo group ($p < 0.07$) (Fig 3).

Similar to *in vivo* PET with ATSM, cardiac PBR28 levels increased one week post-intoxication and returned to baseline values by the time of tissue collection [26]. 12 week PBR28 data significantly correlated across cardiac regions and levels with perivascular HLA-DR-ir ratings ($r_{rm} = 0.23$; $p < 0.018$), but not with nerve bundle HLA-DR-ir ($r_{rm} = 0.15$; $p > 0.11$) (Fig 3). When tests for correlation between 12 week PBR28 and perivascular HLA-DR-ir ratings were performed for each treatment group individually, the correlation remained significant for the 6-OHDA + placebo group, but not for the 6-OHDA + pioglitazone group (S12 Fig; S7 Table).

## Expression of markers of PPARγ activation

PGC1α-ir and CD36-ir were analyzed to assess PPARγ target engagement by pioglitazone. PGC1α was expressed in nearly all myocardial cell types, with notable absence in the interstitial connective tissue surrounding vessels and nerves (S10 Fig). No difference in PGC1α-ir was observed between treatment groups in cardiomyocytes (overall treatment effect: %AAT, $p > 0.71$, $\eta_p^2 = 0.056$; OD, $p > 0.41$, $\eta_p^2 = 0.137$), and a trend was observed only for nerve bundle OD (overall treatment effect: %AAT, $p > 0.11$, $\eta_p^2 = 0.307$; OD, $p < 0.093$, $\eta_p^2 = 0.328$).

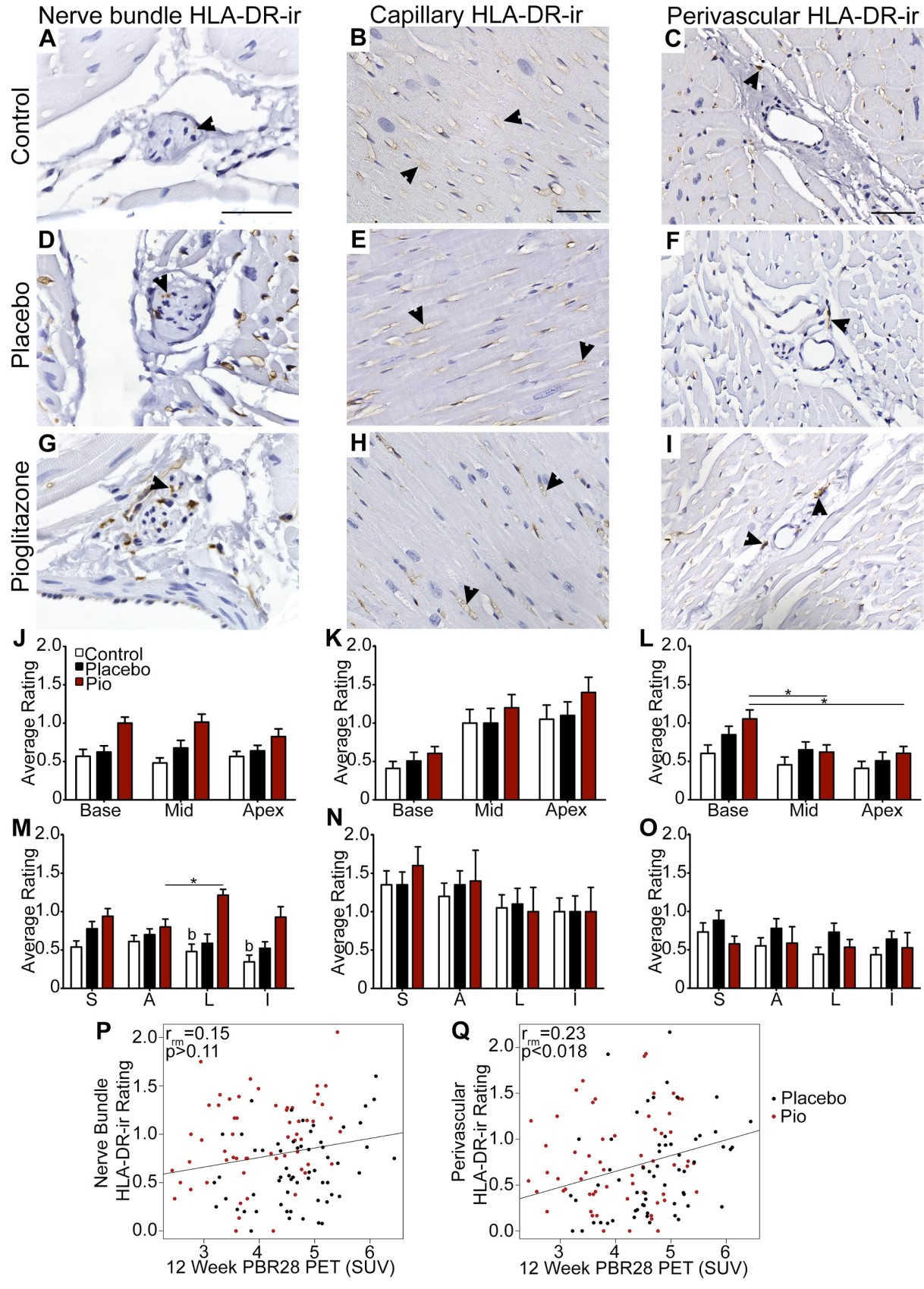

**Fig 3. Cardiac inflammation was mildly affected by treatment and perivascular HLA-DR-ir correlated with *in vivo* data.** (A-I) Photomicrographs of left ventricle (A, D, G) nerve bundles, (B, E, H) capillaries, and (C, F, I) perivascular immune cells immunostained for the antigen presenting cell marker HLA-DR in the (A, B, C) control, (D, E, F) 6-OHDA + placebo, and (G, H, I) 6-OHDA + pioglitazone groups. Scale bar = (A, B, C) 50 μm. Arrowheads point to HLA-DR-ir. Graphs of ratings of (J,M) cardiac nerve bundle, (K,N) capillary, or (L,O) perivascular HLA-DR-ir at cardiac (J,K,L) levels or (M,N,O) regions. Each level is an average of the four regions in that level and each region is an average of the three levels in that region. (P, Q) Plots of repeated measures correlation across cardiac levels and regions of left ventricle for (P) nerve bundle HLA-DR-ir or (Q) perivascular HLA-DR-ir with *in vivo* evaluation of inflammation at 12 week post-6-OHDA as measured by PBR28 PET. Each point represents a single anatomical area in one animal (12 total anatomical areas (3 levels * 4 regions) per animal; 10 animals). Error bars = SEM. b, 6-OHDA + pioglitazone group significantly different from the control group. *, significant difference between two anatomical areas within one treatment group; see text for details. 6-OHDA, 6-hydroxydopamine; HLA-DR-ir, human leukocyte antigen DR immunoreactivity; Pio, pioglitazone; Mid, middle; S, septal; A, anterior; L, lateral; I, inferior; PBR28, [11C]PBR28; PET, positron emission tomography; SUV, standard uptake volume.

CD36-ir was visible at low levels intracellularly within cardiomyocytes and to varying degrees in capillary endothelial cells and cardiomyocyte sarcolemma (Fig 4). Qualitatively, CD36 appeared to have higher expression at the sarcolemma in 6-OHDA + pioglitazone animals compared to other groups, most notably at the intercalated discs, the Z line of the sarcomere where two cardiomyocytes connect. Analysis of CD36-ir %AAT and OD in the middle cardiac level showed a significant overall effect of condition (%AAT, $p < 0.037$, $\eta_p^2 = 0.425$; OD, $p > 0.16$, $\eta_p^2 = 0.264$). Post hoc analysis showed a trend toward significantly higher CD36-ir in cardiomyocytes of 6-OHDA + pioglitazone-treated animals compared to the control group (%AAT, $p < 0.069$; OD, $p > 0.326$) and the 6-OHDA + placebo group (%AAT, $p < 0.078$; OD, $p > 0.266$), and no difference between the 6-OHDA + placebo and 6-OHDA + pioglitazone groups (%AAT, $p = 1$; OD, $p = 1$). The effect of region showed a trend toward significance (%AAT, $p < 0.08$, $\eta_p^2 = 0.175$; OD, $p > 0.186$, $\eta_p^2 = 0.124$). Post hoc tests for individual regions showed that the 6-OHDA + pioglitazone group had higher CD36-ir %AAT than the control group at inferior region ($p < 0.03$), with a trend toward a significant difference at the septal region (%AAT, $p < 0.067$; OD, $p > 0.239$) (Fig 4). Similarly, the 6-OHDA + pioglitazone group had higher CD36-ir %AAT than the 6-OHDA + placebo group at the lateral region ($p < 0.043$), with a trend toward a significant difference at the inferior region (%AAT, $p < 0.08$; OD, 0.458) (Fig 4). In the 6-OHDA + pioglitazone group, the septal region showed significantly higher CD36-ir %AAT than the anterior region ($p < 0.03$); differences between individual regions were not observed in the other treatment groups (Fig 4).

Semiquantitative rating of CD36-ir at cardiomyocyte intercalated disks confirmed a significant overall effect of condition ($p < 0.001$, $\eta_p^2 = 0.699$) with significantly more CD36-ir in 6-OHDA + pioglitazone animals compared to the control ($p < 0.003$) and 6-OHDA + placebo ($p < 0.003$) groups and no difference between the control and 6-OHDA + placebo groups ($p = 1$). The effect of region was also significant ($p < 0.02$, $\eta_p^2 = 0.239$). Post hoc tests for individual regions show that the 6-OHDA + pioglitazone group had higher CD36-ir intercalated disk ratings than the control group at the septal ($p < 0.022$), lateral ($p < 0.001$), and inferior ($p < 0.002$) regions, with a trend toward a significant difference at the anterior region ($p < 0.07$; Fig 4). Similarly, the 6-OHDA + pioglitazone group had higher CD36-ir intercalated disk ratings than the 6-OHDA + placebo group at the lateral ($p < 0.0002$) and inferior ($p < 0.006$) regions, with a trend toward a significant difference in the anterior region ($p < 0.06$). In the 6-OHDA + placebo group, the septal region showed significantly higher CD36-ir intercalated disc rating than the lateral region ($p < 0.012$); differences between individual regions were not observed in the other treatment groups (Fig 4). CD36-ir intercalated disc semiquantitative ratings significantly correlated across cardiac regions with cardiomyocyte CD36-ir %AAT ($r_{rm} = 0.606$, $p < 0.000008$) and OD ($r_{rm} = 0.568$, $p < 0.00004$) (Fig 4). All CD36-ir correlations remained significant when each treatment group was analyzed individually, except the correlation between semiquantitative rating and CD36-ir OD in the control group (S13 Fig; S7 Table).

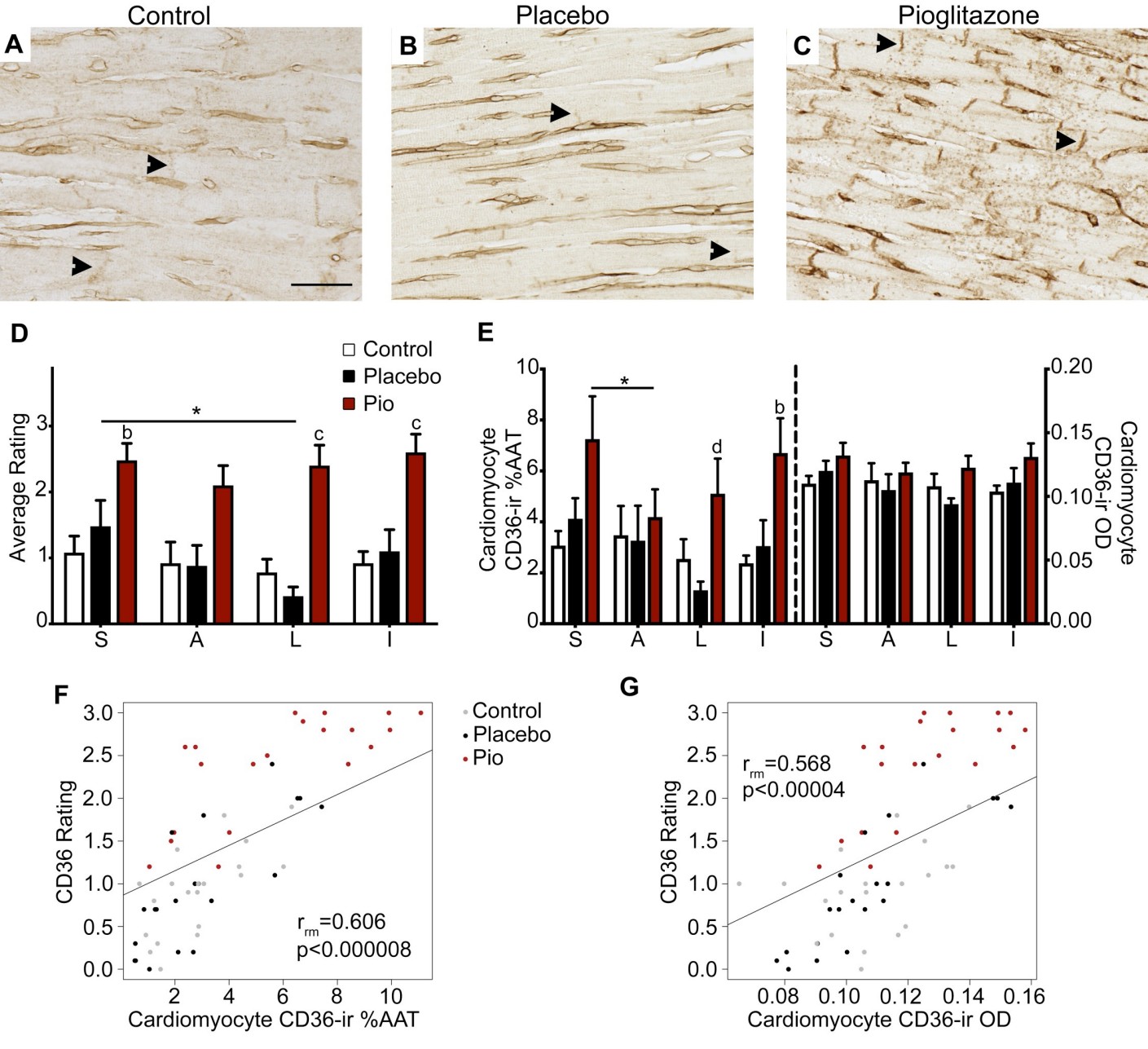

**Fig 4. CD36-ir at cardiomyocyte intercalated discs was significantly increased by PPARγ activation.** (A-C) Photomicrographs of left ventricle cardiomyocytes immunostained for CD36 in the (A) control, (B) 6-OHDA + placebo, and (C) 6-OHDA + pioglitazone groups. Scale bar = 50 μm. Arrowheads point to CD36-ir at intercalated discs. (D) CD36-ir rating at intercalated discs. (E) CD36-ir %AAT and OD in cardiomyocytes. (F, G) Plots of repeated measures correlations across cardiac regions of left ventricle CD36 intercalated disc rating with cardiomyocyte CD36-ir (F) %AAT and (G) OD. Each point represents a single region in one animal (4 regions; 15 animals). Error bars = SEM. b, 6-OHDA + pioglitazone group significantly different from control group; c, 6-OHDA + pioglitazone group significantly different from control and 6-OHDA + placebo group; d, 6-OHDA + pioglitazone group significantly different from 6-OHDA + placebo group. *, significant difference between two anatomical areas within one treatment group; see text for details. 6-OHDA, 6-hydroxydopamine; PPARγ, peroxisome proliferator-activated receptor gamma; CD36-ir, cluster of differentiation 36 immunoreactivity; S, septal; A, anterior; L, lateral; I, inferior; Pio, pioglitazone; %AAT, percent area above threshold; OD, optical density.

## Adrenal medulla

TH-ir and AADC-ir the adrenal medulla were observed in the cytoplasm of medullary chromaffin cells in all monkeys (Fig 5). Adrenal TH-ir and AADC-ir %AAT and OD were significantly affected by treatment (TH: %AAT $p < 0.00006$ ($\eta^2 = 0.802$), OD $p < 0.08$ ($\eta^2 = 0.35$); AADC %AAT $p < 0.002$ ($\eta^2 = 0.649$), OD $p < 0.0008$ ($\eta^2 = 0.695$)). Post hoc analysis indicated that TH-ir (%AAT, $p < 0.00006$; OD, $p > 0.09$) and AADC-ir (%AAT, $p < 0.002$; OD, $p < 0.0007$) were significantly lower in 6-OHDA + placebo animals compared to normal controls (Fig 5). The only significant difference between the 6-OHDA + pioglitazone animals and normal controls was AADC OD ($p < 0.04$) (Fig 5). The 6-OHDA + pioglitazone group also had significantly higher TH-ir %AAT than the 6-OHDA + placebo group (%AAT, $p < 0.002$; OD, $p > 0.29$) (Fig 5).

## Circulating catecholamine levels

Assessment of circulating catecholamine levels by HPLC demonstrated individual variations and changes over time (Fig 6; S6R Table). No significant differences were found between groups at baseline (NE: $p > 0.69$, $r = 0.17$; E: $p = 1$, $r = 0.03$; DA: $p > 0.84$, $r = 0.1$) although DOPAC did show a trend toward a group difference ($p < 0.096$, $r = 0.56$). Norepinephrine values decreased from baseline to one week in a manner that trended toward significance for both the 6-OHDA + placebo and 6-OHDA + pioglitazone groups ($p < 0.07$ and $r = 0.64$ for both) (Fig 6). This percent decrease from baseline to 1 week was statistically significantly greater for the 6-OHDA + placebo group than 6-OHDA + pioglitazone-treated animals ($p < 0.016$; $r = 0.81$) (Fig 6). By 12 weeks post-neurotoxin, norepinephrine levels were similar between groups ($p > 0.19$; $r = 0.51$). E, DA, and DOPAC did not show any significant changes over time or between treatment groups (Fig 6; S6R Table).

## Discussion

The present post mortem study in a rhesus monkey model of PD cardiac dysautonomia aimed to validate *in vivo* cardiac PET markers of sympathetic innervation, oxidative stress and inflammation and evaluate PPARγ target engagement by pioglitazone. Based on this premise, three questions emerge for discussion. First, do the post mortem data match *in vivo* PET results? Second, can target engagement of PPARγ by pioglitazone be detected in the heart? Third, are the data generated in this nonhuman primate model relevant for clinical translation?

### Do the post mortem data correspond with *in vivo* PET results?

As mentioned in the introduction, *in vivo* cardiac PET was performed in the 6-OHDA-treated monkeys with the radioligands MHED, ATSM, and PBR28 to assess sympathetic innervation, oxidative stress, and inflammation, respectively. Refined PET imaging methods helped us reduce the number of animals needed to assess mechanisms of cardiac neurodegeneration. Quantification of radioligand uptake over time identified significantly less inflammation and oxidative stress at one week post-6-OHDA in the hearts of pioglitazone-treated animals compared to placebo. Twelve weeks later, these values returned to near-baseline levels for both groups, while sympathetic innervation was regionally preserved in the pioglitazone-treated subjects. At that time point, cardiac tissues were collected and analyzed for TH, 8-OHdG, and HLA-DR expression. When comparing *in vivo* PET data collected ante mortem (12 weeks post-toxin) with post mortem immunohistochemical analysis of protein expression, significant correlations were found between markers of sympathetic innervation and inflammation but

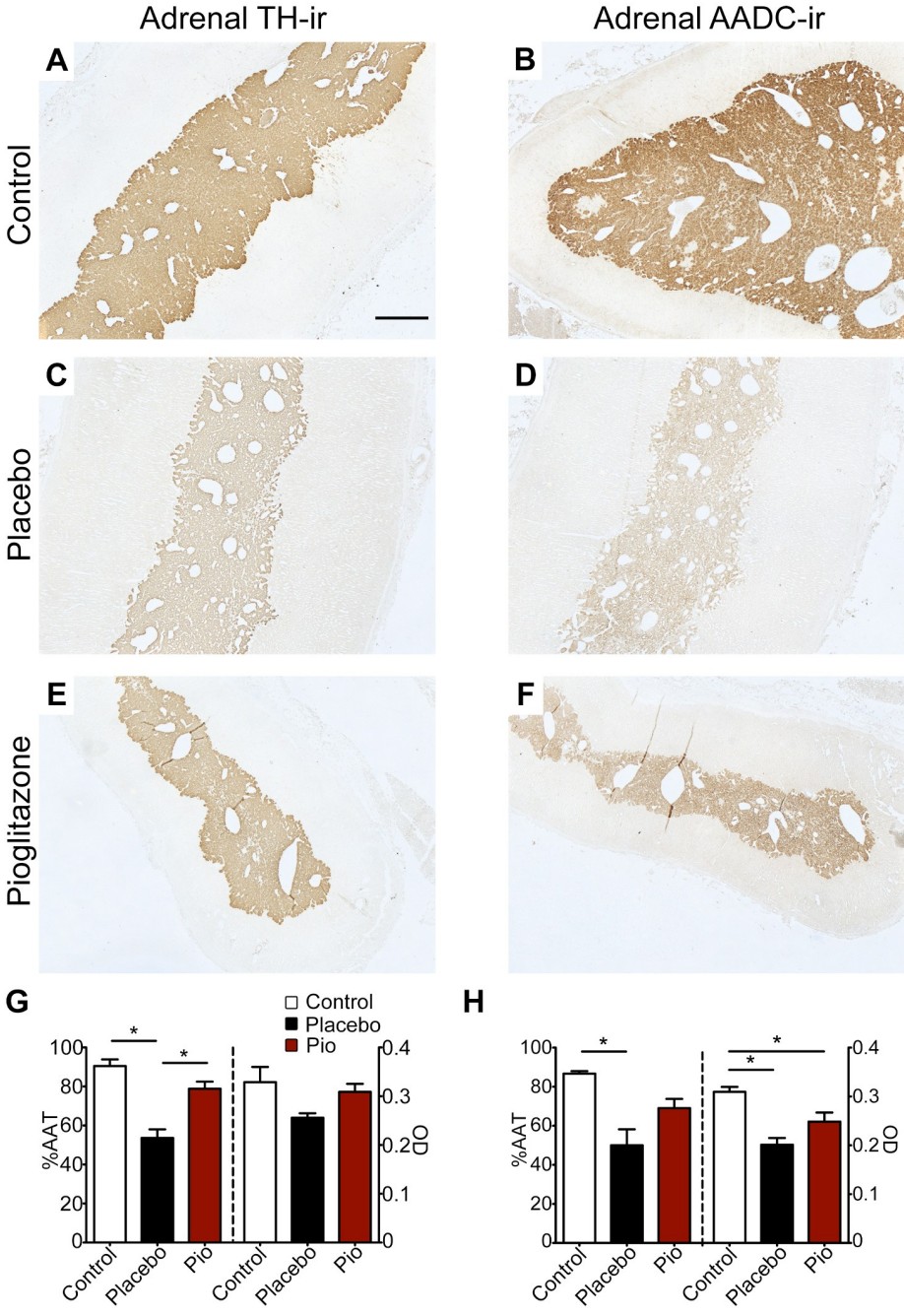

**Fig 5. 6-OHDA-induced TH-ir and AADC-ir loss in the adrenal medulla was attenuated by PPARγ activation.**
(A-F) Photomicrographs of adrenal gland (A, C, E) TH-ir and (B, D, F) AADC-ir in (A, B) control, (C, D) 6-OHDA + placebo, and (E, F) 6-OHDA + pioglitazone groups. Scale bar = 1000μm. Graphs of adrenal medulla (G) TH-ir and (H) AADC-ir. Error bars = SEM. *, statistically significant difference between treatment groups; see text for details. 6-OHDA, 6-hydroxydopamine; PPARγ, peroxisome proliferator-activated receptor gamma; %AAT, percent area above threshold; OD, optical density; TH-ir, tyrosine hydroxylase immunoreactivity; AADC-ir, L-aromatic amino acid decarboxylase immunoreactivity.

not for oxidative stress. Different factors led to these results including timing of data sampling, number of regions analyzed, and specificity of the outcome measure.

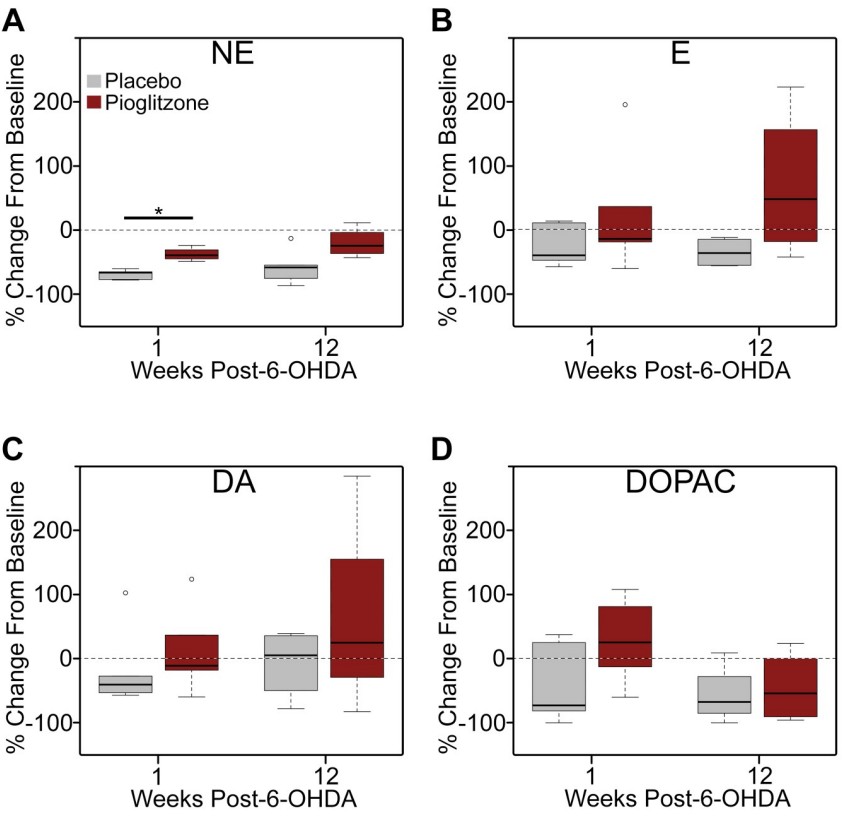

**Fig 6. 6-OHDA-induced loss of circulating norepinephrine 1 week post-neurotoxin was attenuated by PPARγ activation.** (A-D) Box and whisker plots illustrating percent change from baseline to 1 and 12 weeks post-neurotoxin for plasma (A) norepinephrine (NE), (B) epinephrine (E), (C) dopamine (DA), (D) and 3,4 dihydroxyphenylacetic acid (DOPAC). *, p < 0.016. Unfilled circles indicate values that are more than 1.5x the interquartile range above the 75th percentile. 6-OHDA, 6-hydroxydopamine; PPARγ, peroxisome proliferator-activated receptor gamma.

MHED uptake detected by PET at 12 weeks post-6-OHDA showed a significant correlation in its left ventricle distribution with TH-ir in nerve bundles and fibers. MHED is a sympathomimetic amine that is taken up into catecholaminergic neurons by the norepinephrine transporter, packaged into synaptic vesicles, and released into the synapse, allowing visualization of noradrenergic innervation [29]. TH is the enzyme that catalyzes the rate-limiting step in the biosynthesis of catecholamines and is present in catecholaminergic neurons and nerve fibers [30]. To test for a correlation between these two markers in our study, we were able to match the specific regions and levels assessed by TH immunohistochemistry to these same areas in the PET scans. This specificity was due in part to the high resolution of the microPET scanner used for the *in vivo* analysis [26]. In PD patients, cardiac sympathetic denervation is typically evaluated using a different radioimaging marker and technique, metaiodobenzylguanidine (MIBG) scintigraphy [4, 31]. MIBG scintigraphy is preferred in a clinical setting due to its relatively low cost and high availability [32]. However, scintigraphy has poor spatial resolution (15mm) [33] compared to microPET (2mm) [34]. As the cardiac levels analyzed in the present study were 4mm thick, MIBG scintigraphy would not have permitted similarly detailed comparison of *in vivo* and post mortem findings across cardiac anatomy.

Previous research in the systemic 6-OHDA rhesus model illustrated that the cardiac sympathetic lesion induced by 6-OHDA stabilized between 10 and 14 weeks [21, 22]. If a neuroprotective strategy like PPARγ activation by pioglitazone was successful, it would be expected that

preserved cardiac MHED uptake and TH-ir would be present by 12 weeks post-toxin. MHED PET at 12 weeks revealed different distributions of sympathetic innervation in placebo- and pioglitazone-treated animals, suggesting mild preservation of sympathetic innervation at the anterior-lateral region of the base level and the septal region of the apex level associated with pioglitazone [26]. No difference in TH-ir was found between placebo and pioglitazone groups, although TH-ir and MHED uptake correlated across the left ventricle. The dissimilar results from these two techniques may be related to different data analysis approaches in each study. MHED uptake was analyzed by polynomial trend analysis quantifying the pattern of the distribution of sympathetic innervation over 6 cardiac levels and 8 regions [26]. Due to the smaller number of levels (3) and regions (4) evaluated by TH immunohistochemistry, attempts to perform polynomial trend analysis were uninformative. Rather, TH-ir data were analyzed by ANOVA and post hoc tests comparing individual anatomical areas, which were corrected for multiple comparisons. This difference in approach of analyzing *in vivo* and post mortem sympathetic innervation (MHED left ventricle distribution pattern vs. TH-ir in individual anatomical areas) may have impacted the detection of significant differences between groups.

ATSM uptake in the heart returned to baseline values by 12 weeks post-6-OHDA. Similarly, 8-OHdG showed minimal differences between control and toxin-treated animals; the only treatment-related effect was a mild elevation in nerve bundles of the inferior region of 6-OHDA-treated animals compared to controls. ATSM accumulates in electron-rich areas after reduction of the radioactive copper in the radioligand, permitting visualization of oxidative stress [35, 36]. 8-OHdG is an oxidized derivative of deoxyguanosine, and elevated 8-OHdG is a marker of oxidative DNA damage [37]. Previous results in our lab indicate that oxidative protein damage in nerve bundles, as detected by nitrotyrosine-ir, is also at baseline levels by 14 weeks post-6-OHDA [21]. In the current study, the lack of correlation between cardiac 8-OHdG-ir and ATSM might be related to PET data collection being limited to a single ROI in the anterior region of the heart due to high radioligand uptake in the liver obscuring other parts of cardiac anatomy [26]. This restricted PET data collection impacted the statistical power of the calculation of a correlation by reducing the data to a single point per animal. Additionally, because 6-OHDA-induced oxidative stress had returned to baseline levels by 12 weeks, ATSM uptake and 8-OHdG-ir were similar in all animals, making a statistically significant correlation difficult to identify. Future work in which heart tissue is collected earlier post-6-OHDA, such as the 1 week post-toxin timepoint, would enable comparison during ongoing oxidative stress.

Paralleling ATSM, cardiac PBR28 uptake 12 weeks post-6-OHDA was also similar to baseline levels, suggesting that the inflammatory processes were resolving. HLA-DR, a major histocompatibility complex class II cell surface receptor expressed on antigen presenting cells, was present in multiple cell types in the heart, but only perivascular HLA-DR-ir correlated across the left ventricle with PBR28 uptake. It is interesting to note that this correlation only had a small to medium effect size ($r_{rm}$ = 0.23), likely related to the resolution of inflammation by the time of tissue collection, which yielded a dataset with limited spread with which to detect a correlation. Additionally, the correlation was stronger in the placebo group than in the pioglitazone-treated animals, possibly related to increased inflammation in the placebo animals at 1 week post-6-OHDA [26], although group differences were no longer significant at the 12 week time point. PBR28 is a second generation ligand of 18-kDa translocator protein (TSPO), a mitochondrial membrane protein known to significantly increase its expression levels in macrophages upon activation [38]. Therefore, the presence of a correlation between 12 week PBR28 PET and HLA-DR-ir perivascular immune cell rating, but not capillary or nerve bundle rating, is fitting with this radioligand being known to bind to activated immune cells. These data support PBR28 as a marker of cardiac immune cell infiltration and activation. In humans,

a nucleotide polymorphism has been described in the TSPO gene (rs6971) which can produce a 2:1 difference in PBR28 PET signal based on genotype [39, 40]. This polymorphism has not been reported in nonhuman primates, but studies applying PBR28 PET clinically should be aware of this limitation and perform genotyping of subjects.

## Can target engagement of PPARγ by pioglitazone be detected in the heart?

PPARγ is a type II nuclear receptor that works as a transcription factor by forming a heterodimer with retinoid X receptor (RXR), recruiting coactivators or corepressors based on ligand binding, and modulating gene transcription via binding to PPARγ response elements (PPREs) in the promoters of genes [18]. Two genes that have been identified to have promoter PPREs are *PPARGC1A* [41], which encodes the protein PGC1α, and *CD36* [42–44]. These proteins may serve as biomarkers of PPARγ target engagement. In this study, daily dosing of the PPARγ agonist pioglitazone for 12 weeks did not affect the expression of PGC1α, but induced a significant increase in cardiomyocyte CD36, suggesting that other factors differentially modulate PGC1α gene expression after PPARγ activation.

PGC1α is considered a master regulator of mitochondrial biogenesis [45]. Diabetic patients and animal models of diabetes have abnormally low levels of PGC1α mRNA [46, 47]. Pioglitazone administration has been reported to increase PGC1α mRNA expression in the adipose tissue of diabetic patients [48, 49], in the heart and pancreas of diabetic rats [47], and in skeletal muscle of diabetic mice [50] and to increase PGC1α protein expression in skeletal muscle of diabetic mice [50]. However, administration of pioglitazone to non-diabetic PD patients did not affect leukocyte PGC1α mRNA levels [51], and, in wild type mice, it induced either an increase [52] or decrease [53] in liver PGC1α mRNA levels. These findings, combined with our results, suggest that PGC1α may be a more reliable marker of PPARγ activation in diabetic patients and animal models of diabetes than in subjects without metabolic syndrome.

CD36, also known as fatty acid translocase and SR-B2, is a class B scavenger receptor membrane protein that interacts with multiple ligands in numerous cell types [54]. In the heart, CD36 and PPARγ form a positive feedback loop. CD36 at the cell membrane facilitates long-chain fatty acid entry, fatty acids are natural PPARγ ligands and activate PPARγ, and this leads to PPARγ/RXR binding at the PPRE in the CD36 promoter producing increased CD36 expression [55, 56]. In contrast to lowered PGC1α expression associated with diabetes described above, diabetic humans [57] and rodents [58, 59] have elevated CD36 expression. CD36 levels increase in association with changes in metabolic demand or energy availability, such as insulin administration [60, 61] or high-fat diet [60]. CD36 responsiveness to metabolic abnormalities presents a challenge to its use as a marker of pioglitazone target engagement in diseases such as diabetes. Pioglitazone has shown mixed effects on CD36 levels in patients and animal models of diabetes and obesity [59, 62–64]. In non-diabetic, non-obese humans and animal models, pioglitazone has repeatedly been demonstrated to increase CD36 mRNA or protein expression in multiple cell types including skeletal muscle [65], immune cells [66–70], endothelial cells [70], hepatic stellate cells [71], and adipose tissue [72]. A small number of studies have shown no increase in CD36 following pioglitazone in normal human adipose tissue [73], normal rat liver [74], and, interestingly, normal mouse cardiomyocytes [75]. It is currently unclear why the impact of pioglitazone on cardiomyocyte CD36 is different between our study and research carried out in mice [75]. In addition to a possible species difference, it should be noted that the previous publication quantified CD36 mRNA level, in contrast to the current study, which investigated CD36 protein immunoreactivity and subcellular localization. Overall, this research suggests that CD36 protein levels are an effective marker of pioglitazone target engagement in cardiomyocytes of non-diabetic rhesus macaques, and caution should be used

if evaluating CD36 levels as a biomarker of pioglitazone target engagement in subjects with metabolic disorders.

### Are the data generated in this nonhuman primate model of cardiac neurodegeneration relevant for clinical translation?

Cardiac postganglionic sympathetic neurodegeneration is common in PD [14], and imaging evidence of this pathology is now a supportive criterion for clinical PD diagnosis [32]. Additionally, loss of sympathetic nerves in the heart has been suggested to predict PD in at-risk individuals [76]. We have previously demonstrated by *in vivo* PET that 6-OHDA reliably induces cardiac sympathetic nerve loss in rhesus monkeys [22, 26]. Furthermore, the anatomical pattern of this loss is similar between PD [2, 5, 77] and the 6-OHDA nonhuman primate model [22, 26], with the apex more affected than the base and sparing of the anterior region. In PD, this loss becomes more diffuse over time, eventually affecting the entire left ventricle [3]. Interestingly, although TH-ir loss in cardiac nerve bundles also followed this pattern [26], TH-ir fiber density reported here did not show statistically significant differences between individual cardiac levels or regions in 6-OHDA-treated animals. The control group exhibited significantly higher density of TH-ir fibers in the septal region than the inferior region. This pattern is in line with existing data revealing the inferior region to have decreased sympathetic innervation relative to other cardiac regions in normal, healthy human [78–80] and rhesus monkey [21, 26] subjects.

Sympathetic postganglionic neurons and the adrenal medulla are both important sources of circulating norepinephrine. Histological evaluation of the adrenal gland in PD patients revealed decreased catecholamine content and activity of catecholamine producing enzymes [81–83]. PD patients with orthostatic hypotension have reduced plasma norepinephrine compared to patients without this symptom [15, 84], although it is unclear if this is related to decreased production and release by the adrenal medulla or to sympathetic neurodegeneration. In the current study, systemic 6-OHDA decreased adrenal TH and AADC and circulating norepinephrine; these results are similar to previous findings in this model [21, 22], illustrating replicability. In contrast to the decreased norepinephrine following systemic 6-OHDA, plasma levels of epinephrine were not significantly affected, suggesting that the effects of the neurotoxin on circulating norepinephrine may be at least in part due to peripheral sympathetic neurodegeneration beyond the adrenal medulla. Additionally, the small group size and high variability in epinephrine, dopamine, and DOPAC data sets may have made detecting group differences difficult. The impact of PPARγ activity on the adrenal gland has been evaluated in the context of an antitumor effect on adrenocortical cancer [85] and altered adrenocortical steroidogenesis, specifically reduced aldosterone production [86, 87]. To the authors' knowledge, this study is the first to demonstrate an effect of PPARγ activation on the adrenal medulla, showing that pioglitazone preserved the TH- and AADC-positive phenotype of adrenal chromaffin cells following systemic 6-OHDA. Future studies may shed light on the mechanisms by which circulating norepinephrine and epinephrine levels are differentially affected by systemic 6-OHDA and pioglitazone administration.

In PD, cardiac sympathetic neurodegeneration is associated with accumulation and aggregation of α-syn in the nerve fibers in the heart [6]. However, our work in the present experiments showed no alteration in α-syn levels in the systemic 6-OHDA nonhuman primate model, which can be seen as a limitation of this neurotoxin model. A previous study evaluating cardiac nerve bundle α-syn-ir in the same model also showed no global change in α-syn-ir and a decrease in α-syn in the septal cardiac region [21]. 6-OHDA is a dopamine analog, allowing for specific uptake by catecholamine transporters. Neurodegeneration is induced by 6-OHDA

via the generation of reactive oxygen species derived from autoxidation and interference with mitochondrial complex I [23]. The contrasting effects of PD and systemic 6-OHDA on cardiac α-syn are likely related to the acute, massive toxic effect of 6-OHDA autoxidation compared to the progressive neurodegeneration characteristic of PD.

It should be noted that a clinical trial evaluating pioglitazone administration to PD patients did not reach significance for the primary outcome measure, attenuation of the change in total Unified Parkinson's Disease Rating Scale (UPDRS) scores over 44 weeks [88]. It is possible that the lack of neuroprotective effect of pioglitazone in the clinical trial may be related to the dependence of diagnosis on motor symptoms, which are not observed until significant neuronal loss (30–50%) [89, 90] has already taken place. Interestingly, diabetic patients treated with glitazones had a 28% decreased risk of developing PD compared to patients taking other medications [91]. Therefore, disruption of neurodegenerative mechanisms earlier in the disease course may be necessary to induce neuroprotection. In the current study, pioglitazone administration was started 24 hours following 6-OHDA. This timing was based on previous findings in our lab that pioglitazone administered for 12 weeks starting 24 hours after intracarotid delivery of the catecholaminergic neurotoxin 1-methyl-4-phenyl-1,2,3,6-tetrahydropyridine (MPTP) in rhesus reduced CD68-ir inflammatory cells, attenuated nigral neuron loss, and preserved dopaminergic nerve fibers in the striatum [19]. The explanation for the difference in PPARγ activation neuroprotective efficacy in the brain and in the heart is currently unknown.

Overall, these results validate *in vivo* PET findings of cardiac sympathetic innervation, oxidative stress, and inflammation and illustrate cardiomyocyte CD36 upregulation as a marker of PPARγ target engagement.

## Supporting information

**S1 Fig. Illustration of cardiac levels and regions analyzed in this study.** Hearts were collected from each monkey following 4% PFA perfusion. Each heart was post-fixed in 4% PFA and dehydrated in 70% ethanol prior to trimming for paraffin embedding. To make each paraffin embedded block of heart tissue, each heart was cut in transverse sections, starting at the apex, to produce eight 4mm slices representing cardiac levels from the apex to the base. The three levels labeled 'Apex', 'Middle', and 'Base' in the figure were used in this study. Once the tissue from each level was mounted onto slides, it was subdivided into four cardiac regions for analysis: S, septal; A, anterior; L, lateral; I, inferior. RV, right ventricle; LV, left ventricle. The arrow indicates that the example cardiac tissue section in the right of the figure is from the middle cardiac level.
(TIF)

**S2 Fig. Example images of cardiomyocyte 8-OHdG-ir rating scale.** (A,B) Photomicrographs of 8-OHdG immunoreactivity (-ir) in left ventricle cardiomyocytes. Scale bar = 200 μm. (A) Represents a rating of 0 with <10% of cardiomyocyte nuclei immunoreactive for 8-OHdG, while (B) represents a rating of 3 with >50% of cardiomyocyte nuclei immunoreactive for 8-OHdG. The rating scale also included a possible rating of 1 when a diffuse light 8-OHdG stain was observed in 10–100% of cardiomyocyte nuclei OR a medium/dark stain was observed in 10–20% of nuclei and a possible rating of 2 when a medium/dark stain was observed in >20% of nuclei. Black arrowheads point to 8-OHdG-ir cardiomyocyte nuclei. 8-OHdG, 8-hydroxy-2'-deoxyguanosine.
(TIF)

**S3 Fig. Example images of rating scales for HLA-DR-ir in capillaries, perivascular immune cells, and nerve bundles.** (A-F) Photomicrographs of left ventricle (A, D) capillaries, (B, E)

perivascular immune cells, and (C, F) nerve bundles immunostained for the antigen present-ing cell marker HLA-DR. Scale bar = 50 μm. (A, B, C) Represent ratings of 0 for each type of HLA-DR immunoreactivity (-ir), while (D, E, F) represent a rating of 3 for each type of HLA-DR-ir. (A, D) Capillary HLA-DR-ir rating is based on the percent area of each cardiac region that exhibited HLA-DR-ir capillaries with a (A) 0 rating having <50% of the region exhibiting HLA-DR-ir capillaries and a (D) 3 rating exhibiting 100% of the region with medium/dark HLA-DR-ir capillaries. The rating scale also included a possible rating of 1 for >50% but less than 100% of the region exhibiting HLA-DR-ir capillaries and a possible rat-ing of 2 for 100% of the region exhibiting light HLA-DR-ir capillaries. (B, E) Perivascular HLA-DR-ir rating is based on the number of HLA-DR-ir cells present within 350 μm of the vessel with a (B) 0 rating having 0 HLA-DR-ir cells and a (E) 3 rating having >10 HLA-DR-ir cells. The rating scale also included a possible rating of 1 for 1 to 5 HLA-DR-ir cells and a pos-sible rating of 2 for 6 to 10 HLA-DR-ir cells. (C, F) Nerve bundle HLA-DR-ir rating is based on the number of HLA-DR-ir cells counted in a nerve bundle with a (C) 0 rating having 0 HLA-DR-ir cells in the bundle and a (F) 3 rating having >5 HLA-DR-ir cells in the bundle. The rating scale also included a possible rating of 1 for 1 or 2 HLA-DR-ir cells in the bundle and a possible rating of 2 for 3 to 5 HLA-DR-ir cells in the bundle. Black arrowheads point to HLA-DR-ir in (D) capillaries, (E) perivascular immune cells, or (F) cells in nerve bundles. HLA-DR, human leukocyte antigen DR.
(TIF)

**S4 Fig. Example images of cardiomyocyte intercalated disc CD36-ir rating scale.** (A,B) Pho-tomicrographs of CD36 immunoreactivity (-ir) in left ventricle cardiomyocytes. Scale bar = 50 μm. (A) Represents a rating of 0 with no to very little CD36-ir at intercalated discs, while (B) represents a rating of 3 with >1/3 of the area of the image showing regularly visible CD36-ir at the intercalated discs. The rating scale also included a possible rating of 1 when CD36-ir was present very lightly in 10–100% of intercalated discs and/or a few (2–4) discs had medium/dark CD36-ir and a possible rating scale of 2 when <1/3 of the area shows regularly visible CD36-ir. Black arrowheads point to CD36-ir intercalated discs. CD36, cluster of differ-entiation 36.
(TIF)

**S5 Fig. Systemic 6-OHDA and PPARγ activation by pioglitazone did not affect cardiac and adrenal medulla tissue architecture.** (A-C) Photomicrographs of cardiac left ventricle myo-cardial nerve bundles, a collection of nerve fibers surrounded by an epineurium, at 63x stained with HE in (A) control, (B) 6-OHDA + placebo, and (C) 6-OHDA + pioglitazone groups. (D-F) Photomicrographs of the adrenal gland at 2.5x stained with HE with medulla visible in the center of the section in (D) control, (E) 6-OHDA + placebo, and (F) 6-OHDA + pioglita-zone groups. Scale bar = (A) 50 μm or (D) 2000 μm. 6-OHDA, 6-hydroxydopamine; PPARγ, peroxisome proliferator-activated receptor gamma; HE, hematoxylin and eosin.
(TIF)

**S6 Fig. α-Synuclein expression was not significantly affected by 6-OHDA or PPARγ activa-tion.** (A-C) Photomicrographs of cardiac left ventricle nerve bundles with α-synuclein (α-syn) immunoreactivity (-ir) in the (A) control, (B) 6-OHDA + placebo, and (C) 6-OHDA + pioglita-zone groups. Scale bar = 50 μm. (D) Across all cardiac levels, no statistically significant differ-ences in α-syn-ir %AAT or OD in nerve bundles were found between or within treatment groups. (E) Across all cardiac regions, no statistically significant differences in α-syn-ir %AAT or OD in nerve bundles were found between or within treatment groups. Error bars = SEM. 6-OHDA, 6-hydroxydopamine; PPARγ, peroxisome proliferator-activated receptor gamma; %

AAT, percent area above threshold; OD, optical density; Pio, pioglitazone; Mid, middle; S, septal; A, anterior; L, lateral; I, inferior.
(TIF)

**S7 Fig. Correlation between nerve bundle HLA-DR-ir semiquantitative ratings and %AAT and OD.** (A) Across cardiac regions, HLA-DR immunoreactivity (-ir) in nerve bundles, as measured by %AAT and OD, was not statistically significantly different between or within treatment groups. Error bars = SEM. (B,C) Plots of repeated measures correlations between nerve bundle HLA-DR-ir semiquantitative ratings and HLA-DR-ir (B) %AAT and (C) OD across cardiac regions in the base level. (B,C) Each point represents a single region in the base level in one animal (4 regions; 15 animals). HLA-DR; human leukocyte antigen DR; Pio, pioglitazone; %AAT, percent area above threshold; OD, optical density S, septal; A, anterior; L, lateral; I, inferior.
(TIF)

**S8 Fig. No co-labeling of HLA-DR with CD68 was observed in nerve bundles in any treatment group.** (A-I) Representative photomicrographs of cardiac left ventricle nerve bundles with double-label immunofluorescent labeling of HLA-DR and CD68 in the (A-C) control group, (D-F) 6-OHDA + placebo group, and (G-I) 6-OHDA + pioglitazone group. Note that there are no CD68+ cells in nerve bundles in any group. Scale bar = 100 μm. (C, F, I) Dashed circles outline nerve bundles; white arrows indicate HLA-DR-ir inside nerve bundles; white arrowheads indicate cells outside of nerve bundles that co-label for CD68 and HLA-DR-ir. HLA-DR; human leukocyte antigen DR; CD68, cluster of differentiation 68.
(TIF)

**S9 Fig. Minimal co-labeling of HLA-DR with S100B was observed in nerve bundles in all treatment groups.** (A-I) Representative photomicrographs of cardiac left ventricle nerve bundles with double-label immunofluorescent labeling of HLA-DR and S100B in the (A-C) control group, (D-F) 6-OHDA + placebo group, and (G-I) 6-OHDA + pioglitazone group. Note minimal HLA-DR/S100B co-labeling in nerve bundles (white arrowheads). Scale bar = 100 μm. (C, F, I) Dashed circles outline nerve bundles; white arrows indicate HLA-DR-ir inside nerve bundles; white arrowheads indicate cells inside of nerve bundles that co-label for S100B and HLA-DR. HLA-DR; human leukocyte antigen DR; S100B, S100 calcium-binding protein B.
(TIF)

**S10 Fig. Systemic 6-OHDA and PPARγ activation had no effect on PGC1α-ir.** (A-F) Photomicrographs of left ventricle (A, C, E) cardiomyocytes and (B, D, F) nerve bundles immunostained for PGC1α in (A, B) control, (C, D) 6-OHDA + placebo, and (E, F) 6-OHDA + pioglitazone groups. Scale bar = (A, B) 50 μm. (G, H) No differences between or within treatment groups were found for (G) cardiomyocyte PGC1α immunoreactivity (-ir) %AAT or OD or for (H) nerve bundle PGC1α-ir %AAT or OD. Error bars = SEM. 6-hydroxydopamine; PGC1α, peroxisome proliferator-activated receptor gamma (PPARγ) coactivator 1-alpha; Pio, pioglitazone; %AAT, percent area above threshold; OD, optical density; S, septal; A, anterior; L, lateral; I, inferior.
(TIF)

**S11 Fig. TH-ir fiber density repeated measures correlations for individual treatment groups.** Plots of repeated measures correlations for TH-ir fiber density with (A-C) nerve bundle TH-ir %AAT, (D-F) nerve bundle TH-ir OD, and (G,H) 12 week MHED PET in (A, D, G) 6-OHDA + placebo-treated, (B, E, H) 6-OHDA + pioglitazone-treated, and (C,F) control

animals. The dot and line color in each graph represent one animal; note that the same colors are used across treatment groups although these represent different animals in each treatment group. 6-OHDA, 6-hydroxydopamine; TH, tyrosine hydroxylase; %AAT, percent area above threshold; OD, optical density; MHED, [11C]meta-hydroxyephedrine.
(TIF)

**S12 Fig. HLA-DR-ir repeated measures correlations for individual treatment groups.** Plots of repeated measures correlations for HLA-DR perivascular semiquantitative rating with 12 week PBR28 uptake in (A) 6-OHDA + placebo- and (B) 6-OHDA + pioglitazone-treated animals. The dot and line color in each graph represent one animal; note that the same colors are used across treatment groups although these represent different animals in each treatment group. 6-OHDA, 6-hydroxydopamine; HLA-DR, human leukocyte antigen DR; PBR28, [11C] PBR28.
(TIF)

**S13 Fig. CD36-ir repeated measures correlations for individual treatment groups.** Plots of repeated measures correlations for CD36 rating with (A-C) cardiomyocyte CD36-ir %AAT or (D-F) cardiomyocyte CD36-ir OD in (A,D) 6-OHDA + placebo-treated, (B,E) 6-OHDA + pioglitazone-treated, or (C,F) control animals. The dot and line color in each graph represent one animal; note that the same colors are used across treatment groups although these represent different animals in each treatment group. 6-OHDA, 6-hydroxydopamine; CD36, cluster of differentiation 36; %AAT, percent area above threshold; OD, optical density.
(TIF)

**S1 Table. Information about rhesus macaques (*Macaca mulatta*) used in the study.**
(DOCX)

**S2 Table. Primary antibodies used for brightfield immunohistochemistry.**
(DOCX)

**S3 Table. Primary antibodies used for double-label immunofluorescence.**
(DOCX)

**S4 Table. Cardiac tissue histological description.**
(DOCX)

**S5 Table. Adrenal tissue histological description.**
(DOCX)

**S6 Table. Data used for statistical analysis.** Each tab of this table contains the data used for the analysis of: (A) TH fiber density correlated with TH nerve bundle immunoreactivity, (B) TH fiber density correlated with MHED uptake, (C) α-syn, (D) 8-OHdG in nerve bundles, (E) 8-OHdG in cardiomyocytes, (F) 8-OHdG correlated with ATSM uptake, (G) HLA-DR capillary rating, (H) HLA-DR perivascular rating, (I) HLA-DR nerve bundle rating, (J) HLA-DR ratings correlated with PBR28 uptake, (K) HLA-DR ImageJ data correlated with PBR28 uptake, (L) HLA-DR nerve bundle ImageJ, (M) PGC1α in nerve bundles, (N) PGC1α in cardiomyocytes, (O) CD36, (P) adrenal TH, (Q) adrenal AADC, or (R) HPLC for circulating catecholamines.
(XLSX)

**S7 Table. Confidence intervals of individual treatment group repeated measures correlations.** For each individual treatment group repeated measures correlation shown in S11–S13 Figs, this table gives the p value, analytic 95% confidence interval (assumes a normal

distribution), and bootstrap confidence interval (does not assume a normal distribution) for the test of correlation.
(XLSX)

**S8 Table. ARRIVE guidelines checklist.**
(PDF)

## Acknowledgments

The authors gratefully acknowledge Gabriel Wachowski, Henry Resnikoff, Dane Shank, Carissa Boettcher, Dr. Amita Kapoor, Dr. Toni Ziegler, Dr. Kevin Brunner, and the dedicated animal care and veterinary staff at the Wisconsin National Primate Research Center for their technical support.

## Author Contributions

**Conceptualization:** Jeanette M. Metzger, Marina E. Emborg.

**Data curation:** Jeanette M. Metzger, Marina E. Emborg.

**Formal analysis:** Jeanette M. Metzger, Colleen F. Moore.

**Funding acquisition:** Marina E. Emborg.

**Investigation:** Jeanette M. Metzger, Helen N. Matsoff, Alexandra D. Zinnen, Rachel A. Fleddermann, Viktoriya Bondarenko, Heather A. Simmons, Andres Mejia, Colleen F. Moore, Marina E. Emborg.

**Methodology:** Jeanette M. Metzger, Alexandra D. Zinnen, Viktoriya Bondarenko, Colleen F. Moore, Marina E. Emborg.

**Project administration:** Jeanette M. Metzger, Viktoriya Bondarenko, Marina E. Emborg.

**Resources:** Marina E. Emborg.

**Software:** Jeanette M. Metzger, Colleen F. Moore.

**Supervision:** Marina E. Emborg.

**Validation:** Jeanette M. Metzger, Viktoriya Bondarenko, Marina E. Emborg.

**Visualization:** Jeanette M. Metzger, Helen N. Matsoff, Alexandra D. Zinnen, Rachel A. Fleddermann.

**Writing – original draft:** Jeanette M. Metzger, Marina E. Emborg.

**Writing – review & editing:** Jeanette M. Metzger, Helen N. Matsoff, Alexandra D. Zinnen, Rachel A. Fleddermann, Viktoriya Bondarenko, Heather A. Simmons, Andres Mejia, Colleen F. Moore, Marina E. Emborg.

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
