## [Decision Letter · Decision Letter 0]

13 Aug 2019

PONE-D-19-17539

Neuroanatomical effects of PPARγ activation in a nonhuman primate model of cardiac sympathetic neurodegeneration

PLOS ONE

Dear Dr Emborg,

Thank you for submitting your manuscript to PLOS ONE. After careful consideration, we feel that it has merit but does not fully meet PLOS ONE’s publication criteria as it currently stands. Therefore, we invite you to submit a revised version of the manuscript that addresses the points raised during the review process.

We would appreciate receiving your revised manuscript by Sep 27 2019 11:59PM. To enhance the reproducibility of your results, we recommend that if applicable you deposit your laboratory protocols in protocols.io, where a protocol can be assigned its own identifier (DOI) such that it can be cited independently in the future. For instructions see: http://journals.plos.org/plosone/s/submission-guidelines#loc-laboratory-protocols

We look forward to receiving your revised manuscript.

Kind regards,

Michael Bader

Academic Editor

PLOS ONE

Journal Requirements:

2. As part of your revision, please complete and submit a copy of the ARRIVE Guidelines checklist, a document that aims to improve experimental reporting and reproducibility of animal studies for purposes of post-publication data analysis and reproducibility: https://www.nc3rs.org.uk/arrive-guidelines. Please include your completed checklist as a Supporting Information file. Note that if your paper is accepted for publication, this checklist will be published as part of your article.

Reviewers' comments:

Reviewer's Responses to Questions

**Comments to the Author**

1. Is the manuscript technically sound, and do the data support the conclusions?

Reviewer #1: Yes

Reviewer #2: Partly

2. Has the statistical analysis been performed appropriately and rigorously? 

Reviewer #1: Yes

Reviewer #2: No

3. Have the authors made all data underlying the findings in their manuscript fully available?

Reviewer #1: Yes

Reviewer #2: Yes

4. Is the manuscript presented in an intelligible fashion and written in standard English?

Reviewer #1: Yes

Reviewer #2: Yes

5. Review Comments to the Author

Reviewer #1: Neuroanatomical effects of PPARγ activation in a nonhuman primate model of cardiac sympathetic neurodegeneration

The dysautonomia’s model proposed by the authors is the systemic 6OHDA injection and its effects upon cardiac cathechoalminergic inervation and adrenal medulla cromaffin cell integrity. Oxidative stress investigated and inflamation as trigher of cellular loss was investigated. Neuroprotective effect of pioglitazone an anti-diabetic drug was also analysed.

The aim of the present study is relevant to the Parkinson’s disease physiopathology. Parkinsonism is much more a syndrome than a disease. The methodology is in accordance with the aim of the study, statistical analysis utilized very robust and the results can help to shed light to early parkinsonism diagnosys and treatment of non motor effects.

I have one question to the authors, which does not invalidade the presente model, my question is:

Considering that parkinsonism has its damage in substantia nigra, a mesencephalic structure, with death of dopaminergic mesostriatal fibres. The cause or trigger this cellular death is still unkwon from its description, 1817 till now. In the model presented in your paper, cathechoalminergic impairment is caused by systemic 6OHDA injection, which pathway in a parkinsoniam patient you see similarities? Are your model suggesting that maybe parkinsonism does not have as main inicial target dopaminergic nigral cells or saying in another way could the disease or syndrome starts in cathechoalmiergic peripheral terminal and progresses to central neurons? If so which intermediate of cathechoalmiergic catabolism could be the vilan?

Reviewer #2: Metzger and colleagues performed a post mortem examination of heart and adrenal tissue in rhesus macaques challenged with the catecholaminergic neurotoxin 6-hydroxydopamine (6-OHDA) and treated them with pioglitazone, a peroxisome proliferator-activated receptor gamma (PPARgamma) agonist, or placebo. These animals were compared to age- and sex-matched normal macaques in order to validate the in vivo cardiac positron emission tomography (PET) findings (which have been already reported in a previous study) and further assess the effects of PPARgamma activation. In summary, Metzger and colleagues reported that they could validate the in vivo PET findings of cardiac sympathetic innervation, oxidative stress, and inflammation. Moreover, the authors suggested the use of cardiomyocyte CD36 upregulation as a marker of PPARgamma target engagement.

The topic addressed in the manuscript will draw the attention of researchers in the fields of neurodegenerative diseases and cardiac dysautonomia. The present study is a follow-up investigation of a study that was published last year by the same research group in the npj Parkinson’s Disease journal. Although the data included in both manuscripts were collected from the same macaques, I have the impression that the present manuscript tells a different story than the 2018 manuscript does; thus, the present findings may contribute to advance the understanding about the topic. Overall, the manuscript is well written, and the results were displayed in good (i.e., well prepared) figures and tables. The methods are appropriate to answer the questions raised by the authors. However, it is questionable whether the present findings support conclusions. For example, the conclusions were based on Pearson’s correlation coefficients ranging from 0.477 to 0.606. Moreover, there are several results that were reported as statistical trends. This and other concerns are listed as follows:

Major points:

1. The abstract thoroughly describes the findings from a previous manuscript published by the same research group (lines 26 to 35). The abstract should be amended to highlight novel findings and reduce the amount of previously published information.

2. In the original submission, the introduction section is short, while the discussion section is too long; in fact, some information presented in discussion refers to introductory information. Please amend the introduction section to make it more informative (e.g., explain the effects of pioglitazone) and then shorten the discussion.

3. Materials and methods, lines 111 to 119. Please include information about the placebo used and the duration of pioglitazone (or placebo) treatment.

4. Materials and methods, lines 333 to 336. Please describe the statistical analysis used to verify data normality.

5. Materials and methods, lines 368 to 370. Why were the HPLC data normalized before being compared between groups? Were there differences in baseline concentrations of circulating monoamines? The authors should present these baseline values in the revised manuscript.

6. For obvious reasons, the number of animals used in the experiments was low (n = 5 for each group). In many analyses, the p value was close to reach statistical significance and, therefore, authors reported their findings as trends (e.g., lines 462, 529 to 530, 541 to 542). In this context, what was the criterion to classify a given result as a trend? This information should be included in the manuscript, because a p = 0.08 was considered a trend (lines 601 to 604), whereas a p = 0.09 was not (lines 583 to 586).

6.1. Additionally, because of the low number of animals used, the authors should consider presenting effect sizes of the analyses reported in the manuscript.

7. The fact that correlations were performed using pooled data from the three experimental groups could make results artificial (and somehow obvious). Please explain why these correlations were performed using pooled data instead of performing separate correlations with data taken from each group.

8. The conclusions of the manuscript were based on correlation analyses. However, most analyses revealed low Pearson’s correlation coefficients (i.e., “r” values ranging from 0.477 to 0.606). In this sense, do present data support conclusions? Can “r” values lower than 0.61 be considered of clinical relevance? The authors should, at least, use an effect size measure to classify the strength of the correlations.

Minor points:

1. The authors may consider changing the title of the manuscript. Overall, the manuscript is focused on the association between the findings of cardiac sympathetic innervation, oxidative stress, and inflammation obtained using PET and immunohistochemistry rather than on the neuroanatomical effects of PPARgamma activation.

2. Introduction, line 57. The use of the abbreviation “OH” is unnecessary, because the abbreviation was only used once (i.e., line 59) throughout the manuscript.

3. Materials and methods, line 184. Please insert the degree symbol in the following sentence: “overnight at 4C in primary”.

4. Materials and methods, line 305. Please replace the phrase “Expression of TH and AADC were each …” with “Expressions of TH and AADC were each …”.

5. In general, the figure legends are long and repeat information that has already been presented in the text. Please shorten these legends, particularly by decreasing the amount of information related to statistics.

6. Legends of figures 1, 2, 3 and 4. These figures contain several micrographs with arrows. Please indicate in the legends what these arrows are showing.

6. PLOS authors have the option to publish the peer review history of their article (what does this mean?). If published, this will include your full peer review and any attached files.

Reviewer #1: No

Reviewer #2: Yes: Samuel Penna Wanner

---

## [Author Response · Author response to Decision Letter 0]

31 Oct 2019

The responses below are duplicated from the file named "ResponseToReviewers_Metzgeretal_10312019" which was also uploaded as a separate file.

“Reviewer #1: Neuroanatomical effects of PPARγ activation in a nonhuman primate model of cardiac sympathetic neurodegeneration

The dysautonomia’s model proposed by the authors is the systemic 6OHDA injection and its effects upon cardiac cathechoalminergic inervation and adrenal medulla cromaffin cell integrity. Oxidative stress investigated and inflamation as trigher of cellular loss was investigated. Neuroprotective effect of pioglitazone an anti-diabetic drug was also analysed.

The aim of the present study is relevant to the Parkinson’s disease physiopathology. Parkinsonism is much more a syndrome than a disease. The methodology is in accordance with the aim of the study, statistical analysis utilized very robust and the results can help to shed light to early parkinsonism diagnosys and treatment of non motor effects.”

<< We thank the reviewer for his/her appreciation of the experimental and statistical methods used for this study. >>

“I have one question to the authors, which does not invalidade the presente model, my question is: Considering that parkinsonism has its damage in substantia nigra, a mesencephalic structure, with death of dopaminergic mesostriatal fibres. The cause or trigger this cellular death is still unkwon from its description, 1817 till now. In the model presented in your paper, cathechoalminergic impairment is caused by systemic 6OHDA injection, which pathway in a parkinsoniam patient you see similarities? Are your model suggesting that maybe parkinsonism does not have as main inicial target dopaminergic nigral cells or saying in another way could the disease or syndrome starts in cathechoalmiergic peripheral terminal and progresses to central neurons? If so which intermediate of cathechoalmiergic catabolism could be the vilan?”

<<Shared neurodegenerative mechanisms between Parkinson’s disease (PD) and the nonhuman primate (NHP) cardiac sympathetic loss model are oxidative stress and inflammation. 

The neurotoxin 6-OHDA used to induce the NHP model is a dopamine analogue specifically uptaken by catecholamine transporters (Rodriquez-Pallares et al 2007; Blum et al 2001). Inside neurons, 6-OHDA autoxidizes and interferes with mitochondrial complex I leading to accumulation of reactive oxygen species (Rodriquez-Pallares et al 2007; Blum et al 2001). This neurodegeneration is then amplified by immune cell activation and recruitment, leading to further cytokine release, oxidative stress, and neuron death. 

The 3,4-dihydroxy phenylacetaldehyde (DOPAL) is a toxic dopamine metabolite responsible for the high oxidative stress present in nigral dopaminergic neurons. Toxic catecholamine metabolites like DOPAL have been suggested to be the ‘villains’ contributing to central and cardiac catecholaminergic PD neurodegeneration that the reviewer asks about (Goldstein and Sharabi 2019). The postganglionic sympathetic cardiac nerve fibers that are lost in PD are noradrenergic, and are therefore also susceptible to DOPAL toxicity (Goldstein and Sharabi 2019). Oxidative damage leads to the recruitment of inflammatory cells which further contribute to neurodegeneration in PD (Rocha et al 2015; Hirsch et al 1998; McGeer et al 2001; Banati et al 1998; Imamura et al 2003).

We have added additional background information on these mechanisms to the introduction to clarify the shared neurodegenerative mechanisms between PD and the NHP model.

As noted in the discussion section, an additional neurodegenerative mechanism of PD is protein aggregation, specifically accumulation and aggregation of alpha-synuclein. The present NHP cardiac sympathetic loss model does not lead to changes in alpha-synuclein, which we discuss in the “Are the data generated in this nonhuman primate model of cardiac neurodegeneration relevant for clinical translation?” section of the discussion. 

This issue relates to your question of whether the present study suggests that PD neurodegeneration could start outside of the central nervous system. Please note that the goal of the study was to specifically model sympathetic neurodegeneration in the heart. The model in this manuscript is generated by intravenous 6-OHDA. As this neurotoxin does not cross the blood brain barrier (Blum et al 2001), it cannot model PD-like neurodegeneration in the central nervous system (Joers et al 2014). Progression of PD pathology from the periphery to the central nervous system is hypothesized to be related to alpha-synuclein misfolding and aggregation (Olanow and Brundin 2013; Brundin and Melki 2017; Vargas et al 2019). No changes in alpha-synuclein are observed in this model (see S6 Fig and Discussion section “Are the data generated in this nonhuman primate model of cardiac neurodegeneration relevant for clinical translation?” paragraph 3). >>

“Reviewer #2: Metzger and colleagues performed a post mortem examination of heart and adrenal tissue in rhesus macaques challenged with the catecholaminergic neurotoxin 6-hydroxydopamine (6-OHDA) and treated them with pioglitazone, a peroxisome proliferator-activated receptor gamma (PPARgamma) agonist, or placebo. These animals were compared to age- and sex-matched normal macaques in order to validate the in vivo cardiac positron emission tomography (PET) findings (which have been already reported in a previous study) and further assess the effects of PPARgamma activation. In summary, Metzger and colleagues reported that they could validate the in vivo PET findings of cardiac sympathetic innervation, oxidative stress, and inflammation. Moreover, the authors suggested the use of cardiomyocyte CD36 upregulation as a marker of PPARgamma target engagement.

The topic addressed in the manuscript will draw the attention of researchers in the fields of neurodegenerative diseases and cardiac dysautonomia. The present study is a follow-up investigation of a study that was published last year by the same research group in the npj Parkinson’s Disease journal. Although the data included in both manuscripts were collected from the same macaques, I have the impression that the present manuscript tells a different story than the 2018 manuscript does; thus, the present findings may contribute to advance the understanding about the topic. Overall, the manuscript is well written, and the results were displayed in good (i.e., well prepared) figures and tables. The methods are appropriate to answer the questions raised by the authors. However, it is questionable whether the present findings support conclusions. For example, the conclusions were based on Pearson’s correlation coefficients ranging from 0.477 to 0.606. Moreover, there are several results that were reported as statistical trends. This and other concerns are listed as follows:”

 << We thank the reviewer for his/her appreciation of the manuscript and the contribution of the findings to the field. The conclusions made in this manuscript are based entirely on findings presented in the results section and data analyzed with appropriate statistical techniques. 

The reviewer gives the example of Pearson’s correlation coefficients ranging from 0.477 to 0.606. It is important to note that these values, 0.477 and 0.606, are not Pearson’s correlation coefficients; rather, these are repeated measure correlation coefficients (see Methods section “Statistical Analysis”); for more information on this statistical technique, please see methods and the original publication by Bakdash et al (Bakdash, J. Z. & Marusich, L. R. Repeated measures correlation. Front. Psychol. 8, 456 (2017)). Briefly, repeated measures correlations are appropriate when there are multiple observations on each subject such that use of the standard Pearson correlation is inappropriate. For the present case, the 'rmcorr' package in R fits a linear set of relations between quantitative variables that allow the intercepts to vary across animals while holding the slope constant. If the slopes are heterogeneous across animals, the relations are nonlinear, or are near zero, it would be reflected in small effect sizes (i.e., small rrm values, and large confidence intervals, yielding nonsignificance). We have modified wording in the manuscript to clarify that these are repeated measures correlations. In the ‘Statistical analysis’ section of the methods we have added a brief explanation of what repeated measures correlations are. 

It is also important to clarify that for tests of correlation, the correlation coefficient itself represents the effect size. For correlations, the percentage of variance in one variable accounted for by the linear relationship to another variable is estimated as the squared correlation coefficient. Thus, the results of repeated measures correlation coefficients of 0.477 and 0.606 account for 23% and 37% of the variance in the dataset. Additionally, effect sizes of 0.477 and 0.606 are considered “large” effect sizes. As proposed by Cohen (1988, Statistical Power Analysis for the Behavioral Sciences, 2nd ed) and widely repeated and accepted (see Murphy et al., 2009, Statistical Power Analysis, p. 38 ff), conventions for effect sizes of correlations as "small," "medium," and "large," are approximately r = 0.10, 0.30, and 0.50, respectively. Finally, the significance of these effects was also tested and presented as p values; the p value for the test with the coefficient of 0.477 was p<0.0000002 and the p value for the test with the coefficient of 0.606 was p<0.000008. These p values are well below the typical cutoff for statistical significance of 0.05. For the repeated measures correlations, we calculated the confidence intervals both analytically (assuming the normal distribution) and via a bootstrap option (which does not assume the normal distribution) in the ‘rmcorr’ package. The bootstrap values are very close to the analytical values (see below in point #4 for details).

We address the reviewer comment on reporting statistical trends below in our response to major point #6. 

We do wish to note here that during the process of reviewing our data, two copy/paste errors were found that lead to updated effect size and p values for the correlations between post mortem methods of evaluated sympathetic innervation: fiber density (TH fiber density) and TH-ir bundle %AAT and OD. These values have been corrected in the text and in the figures. The below table summarizes this change:

Correlation Name Old Corr. Coeff. Old p-value New Corr. Coeff. New p-value

TH-ir fiber density with bundle TH-ir %AAT 0.503 p<0.00000002 0.399 p<0.0000001

TH-ir fiber density with bundle TH-ir OD 0.477 p<0.0000002 0.384 p<0.0000003

The above and below responses continue to use the examples from the reviewer of effect sizes of 0.477 (which is now updated to 0.384 in the text) and 0.606 (this value is not changed) for consistency. >> 

“Major points:

1. The abstract thoroughly describes the findings from a previous manuscript published by the same research group (lines 26 to 35). The abstract should be amended to highlight novel findings and reduce the amount of previously published information.”

<< Thank you for the comment. We have updated the abstract to include more information on novel findings in the present manuscript following the reviewer’s suggestion. >>

“2. In the original submission, the introduction section is short, while the discussion section is too long; in fact, some information presented in discussion refers to introductory information. Please amend the introduction section to make it more informative (e.g., explain the effects of pioglitazone) and then shorten the discussion.”

<< Following the reviewer’s suggestion, we have added background information to the introduction including moving information about pioglitazone from the discussion to the introduction. >> 

“3. Materials and methods, lines 111 to 119. Please include information about the placebo used and the duration of pioglitazone (or placebo) treatment.”

<< Thanks for the recommendation. We have added information about the placebo used and clarified the duration of the placebo and pioglitazone dosing.>> 

“4. Materials and methods, lines 333 to 336. Please describe the statistical analysis used to verify data normality.”

<< Tests to verify normality were not performed for all data sets, as normality is not necessary for all the statistical techniques performed for this data analysis (Mann-Whitney, Wilcoxon Signed Rank, Spearman correlations, and the bootstrap confidence intervals of the repeated measures correlations). Although inferential parametric statistical techniques are sometimes preceded by formal statistical tests of the fit of the samples to the normal distribution, such distributional pre-testing is not itself without controversy (see a brief review in Rochon et al., 2012, BMC Medical Research Methodology, “To tests or not to test…”). We chose to visually examine the data for outliers and reasonable normality using a function in R that plots data versus the quantiles of the normal distribution (‘qqnorm’). We now note this in the manuscript in the ‘Statistical analysis’ section of the Methods.

For analysis of variance (ANOVA), which was used to test for between and within subject differences in this manuscript, inferences about means are considered to be relatively robust with respect to the assumption of normality. This conclusion was drawn early in the use of ANOVA (e.g., see Scheffe, 1959, The Analysis of Variance, p. 334 ff) and is repeated more recently (e.g., see Maxwell & Delaney, 2004, Designing Experiments and Analyzing Data, p.112ff, "ANOVA is generally robust to violations of the normality assumption, in that even when data are non-normal, the actual Type I error rate is usually close to the nominal (i.e., desired) value.")

Additionally, for ANOVA that includes a repeated measures factor (in our case the levels and regions of the heart tissue), an additional assumption is sphericity of variance-covariance matrix. Sources on ANOVA recommend using adjusted p-values that accommodate possible violations of sphericity. As such, all cardiac data sets analyzed by ANOVA were tested with Mauchly’s Test of Sphericity in SPSS. Because not all datasets passed Mauchly’s Test of Sphericity at p < 0.05, we used the Huynh-Feldt adjusted p value for reporting all within-subject effects, as is widely recommended. We have added additional detail to the ‘Statistical analysis’ section of the Methods to clarify the testing and p value adjustment for sphericity.

HPLC data sets were analyzed in R by comparing between treatment groups (Mann-Whitney) and between time points (Wilcoxon signed-rank); these tests are nonparametric and do not assume the data is normally distributed (Marascuilo & Serlin, 1988, Statistical methods for the social and behavior sciences, ch. 18).

For data that was analyzed by standard Pearson correlation (rather than repeated-measures correlation) we now also report the Spearman correlation in the updated manuscript. The Spearman correlation relies on only the rank order of the data; note that the Spearman and Pearson correlation values reported in the manuscript are very similar. 

To understand the extent to which data distributions could have affected the conclusions of the repeated measures correlations, we have inserted here a data table that shows the confidence intervals of the repeated measures correlations both assuming a normal distribution (labeled as ‘analytic 95% CI’) and calculated via a bootstrap method (labeled as ‘bootstrap 95% CI’) that does not assume a normal distribution. Please note that these confidence intervals are very similar using either method, further supporting the validity of the reported results. 

Correlation name analytic 95% CI bootstrap 95% CI

MHED - ThFiber 0.3631325, 0.6417989 0.3757759, 0.6458818

THAat -ThFiber 0.3480776, 0.6315508 0.3397706, 0.6188137

THFiber - ThOD 0.3173492, 0.6103216 0.3325224, 0.6084663

PBR28 12wk -- HLADR perivasc semiq rating 0.03909911, 0.3968092 0.03886163, 0.3670285

CD36 Rating - CD36AAT 0.3764832, 0.764993 0.441628, 0.7233985

CD36 Rating - CD36OD 0.325629, 0.7398377 0.4259931, 0.6941441

>>

“5. Materials and methods, lines 368 to 370. Why were the HPLC data normalized before being compared between groups? Were there differences in baseline concentrations of circulating monoamines? The authors should present these baseline values in the revised manuscript.”

<< To clarify, these data were normalized with respect to baseline by calculating the percent change from baseline: ((later timepoint – baseline)/baseline)*100. The methods section has been updated to clarify how this was calculated. The data were normalized to baseline due to individual animal variability in circulating catecholamines at baseline. Yes, independent 2-group Mann-Whitney U Tests were performed for each catecholamine at baseline; no significant differences were found between groups, so this data was not included in the original manuscript. We have added the results of these tests to the revised manuscript in the ‘Circulating catecholamine levels’ section of the Results. Additionally, the raw baseline values were included in the original submission in supplementary table 6; HPLC data is tab ‘R’. We have added a reference to ‘(S6R Table)’ to the text of the manuscript in the HPLC results section to clarify where the data can be found.>> 

“6. For obvious reasons, the number of animals used in the experiments was low (n = 5 for each group). In many analyses, the p value was close to reach statistical significance and, therefore, authors reported their findings as trends (e.g., lines 462, 529 to 530, 541 to 542). In this context, what was the criterion to classify a given result as a trend? This information should be included in the manuscript, because a p = 0.08 was considered a trend (lines 601 to 604), whereas a p = 0.09 was not (lines 583 to 586).”

<< We thank the reviewer for this question and the opportunity to address how we define a statistical trend. As the reviewer is aware, the concept of using p value cutoffs for defining statistical significance or a statistical trend has received considerable criticism due to these cutoffs being relatively arbitrary values. Indeed, the use of statistical p-value cutoffs of any sort to decide the worthiness of a scientific study for publication (or any other use) is a controversial topic in the history of statistics, including to the present day (see Wasserstein & Lazar, 2016, “The ASA’s Statement on p-Values: Context, Process, and Purpose,” The American Statistician, 70, 129–133.) As stated by McShane & Gal (2017, “Statisical significance and dichotomization of evidence,” Journal of the American Statistical Association, p.886), “…statisticians have noted the 0.05 threshold (or for that matter any other threshold) used to dichotomize results into statistically significant and not statistically significant is arbitrary (Fisher, 1926; Yule and Kendall 1950; Cramer 1955; Cochran 1976; Cowles and Davis 1982) and thus this dichotomization has “no ontological basis” (Rosnow and Rosenthal 1989).”

 However, because we follow the still-conventional cutoff for statistical significance of p < 0.05, we have now included in the manuscript a cutoff for defining a statistical trend. We have defined this cutoff as 0.1, and we have updated relevant portions of the methods (section ‘Statistical analysis’) and results (sections ‘HLA-DR expression’ and ‘Expression of markers of PPARγ activation’). In addition, as described below, we have included effect sizes of the ANOVA results, and confidence intervals for the repeated-measures correlations.>>

“6.1. Additionally, because of the low number of animals used, the authors should consider presenting effect sizes of the analyses reported in the manuscript.”

<< Following the reviewer’s suggestion, we have added to the updated version of the manuscript the effect size as partial eta squared (ηp2) for main ANOVA effects for the cardiac data, and we have added the effect size as eta squared (η2) for the one way ANOVAs performed to analyze the adrenal data. Additionally, we have added the effect size for all Mann Whitney tests reported as r=Z/(sqrt(N)). We have also updated the methods section stating what effect sizes were reported. As discussed in detail earlier in this letter (pg 3), the correlation coefficient represents the effect size for tests of statistical correlations, and this correlation coefficient is reported for each test of correlation in the previous version of the manuscript. >> 

“7. The fact that correlations were performed using pooled data from the three experimental groups could make results artificial (and somehow obvious). Please explain why these correlations were performed using pooled data instead of performing separate correlations with data taken from each group.”

<< The correlations were performed using pooled data because the goals of the analyses were to understand if the results of detecting oxidative stress, inflammation, sympathetic fiber loss, PPARgamma activation, etc correlated across methods of analysis. This is important for understanding if these different methods show repeatable results, supporting the robustness of the techniques. Thus, the correlation analysis is improved by including data from different treatment groups that will be more likely to show different levels of oxidative stress, PPARgamma activation, etc in order to probe the sensitivity of the methods and the relationship between different methods of quantifying the effects. Additionally, the graphs of these correlations contain symbols unique to each treatment group, so readers will easily be able to visualize potential group differences in the graphs.

To further address the reviewer’s question in depth, we have performed additional tests to analyze a number of the datasets by individual treatment groups, rather than pooled data. This analysis shows that the repeated measures correlations remain significant for each individual group for most data sets:

1) MHED – ThFiber (all groups show significant correlations)

2) THAat -ThFiber (all groups show significant correlations)

3) THFiber – ThOD (all groups show significant correlations)

4) PBR28 12wk and HLADR perivasc semiq rating (placebo group shows significant correlation)

5) CD36 Rating - CD36AAT (all groups show significant correlations)

6) CD36 Rating and CD36OD (placebo and pioglitazone groups show significant/very near significant correlations (pio p=0.0507))

with two exceptions: 

1) PBR28 12wk and HLADR perivasc semiq rating (no significant correlation in pioglitazone group) 

2) CD36 Rating and CD36OD (no significant correlation in control group) 

The above findings are in line with our previous conclusions, although they do provide additional interesting detail that we report and discuss in the updated manuscript. Please see updated results, discussion, new Supp Figs 11-13, and new Supp Table 7 regarding this data. >> 

“8. The conclusions of the manuscript were based on correlation analyses. However, most analyses revealed low Pearson’s correlation coefficients (i.e., “r” values ranging from 0.477 to 0.606). In this sense, do present data support conclusions? Can “r” values lower than 0.61 be considered of clinical relevance? The authors should, at least, use an effect size measure to classify the strength of the correlations.”

<< We interpret this comment to indicate that the reviewer is concerned that the repeated measures correlation r values of 0.477 and 0.606 do not represent large effect sizes. This has been addressed above in detail. Briefly, effect sizes of 0.477 and 0.606 are considered large effect sizes, and tests for group differences further support the significance of these findings with p values of p<0.0000002 and p<0.000008, respectively.

The reviewer’s question of the clinical relevance of the data is a complex one [see, for example: Kimmelman, J., London, A. J., Ravina, B., Ramsay, T., Bernstein, M., Fine, A., … Emborg, M. E. (2009). Launching invasive, first-in-human trials against Parkinson's disease: ethical considerations. Movement disorders: official journal of the Movement Disorder Society, 24(13), 1893–1901. doi:10.1002/mds.22712]. Note that we directly address the question of clinical relevance in the discussion section titled “Are the data generated in this nonhuman primate model of cardiac neurodegeneration relevant for clinical translation?”. In this section, we openly address some of the draw backs of this model, such as the lack of protein accumulation, and the advantages of the model, such as the similar findings between PD and the 6-OHDA model in the distribution of sympathetic loss in the heart and similar loss of catecholamine content in the adrenal medulla. Regarding whether ‘r’ values lower than 0.61 can be considered clinically relevant- there is no specific p value or r value that ensures that pre-clinical research will directly translate to human patients. Rather, investigators must design experiments with the goal of maintaining relevancy to clinical needs. This experiment was designed to maximize translatability through the use of nonhuman primates and by post mortem evaluation of tissue to further our understanding of PET imaging tools used in clinical studies (for more information on the radioligands and PET findings, please see Metzger et al 2018).>> 

“Minor points:

1. The authors may consider changing the title of the manuscript. Overall, the manuscript is focused on the association between the findings of cardiac sympathetic innervation, oxidative stress, and inflammation obtained using PET and immunohistochemistry rather than on the neuroanatomical effects of PPARgamma activation.”

<< We thank the reviewer for this observation. Following the reviewer’s recommendation, we have updated the title of the manuscript to “Post mortem evaluation of inflammation, oxidative stress, and PPARγ activation in a nonhuman primate model of cardiac sympathetic neurodegeneration”. >>

“2. Introduction, line 57. The use of the abbreviation “OH” is unnecessary, because the abbreviation was only used once (i.e., line 59) throughout the manuscript.”

<< We have removed this abbreviation from the manuscript. >>

“3. Materials and methods, line 184. Please insert the degree symbol in the following sentence: “overnight at 4C in primary”.”

<< We have inserted the degree symbol as suggested.>>

“4. Materials and methods, line 305. Please replace the phrase “Expression of TH and AADC were each …” with “Expressions of TH and AADC were each …”.”

<< We have replaced the phrase as suggested. >> 

“5. In general, the figure legends are long and repeat information that has already been presented in the text. Please shorten these legends, particularly by decreasing the amount of information related to statistics.”

<< We have edited to the figure legends to remove information that is redundant with the text of the manuscript. >> 

“6. Legends of figures 1, 2, 3 and 4. These figures contain several micrographs with arrows. Please indicate in the legends what these arrows are showing.”

<< Thank you for pointing out that this is missing. We have added this information to the figure legends. >>

---

## [Decision Letter · Decision Letter 1]

26 Nov 2019

PONE-D-19-17539R1

Post mortem evaluation of inflammation, oxidative stress, and PPARγ activation in a nonhuman primate model of cardiac sympathetic neurodegeneration

PLOS ONE

Dear Dr Emborg,

Thank you for submitting your manuscript to PLOS ONE. After careful consideration, we feel that it has merit but does not fully meet PLOS ONE’s publication criteria as it currently stands. Therefore, we invite you to submit a revised version of the manuscript that addresses the minor points still raised by reviewer 2,

We would appreciate receiving your revised manuscript by Jan 10 2020 11:59PM. To enhance the reproducibility of your results, we recommend that if applicable you deposit your laboratory protocols in protocols.io, where a protocol can be assigned its own identifier (DOI) such that it can be cited independently in the future. For instructions see: http://journals.plos.org/plosone/s/submission-guidelines#loc-laboratory-protocols

We look forward to receiving your revised manuscript.

Kind regards,

Michael Bader

Academic Editor

PLOS ONE

Reviewers' comments:

Reviewer's Responses to Questions

**Comments to the Author**

1. If the authors have adequately addressed your comments raised in a previous round of review and you feel that this manuscript is now acceptable for publication, you may indicate that here to bypass the “Comments to the Author” section, enter your conflict of interest statement in the “Confidential to Editor” section, and submit your "Accept" recommendation.

Reviewer #2: All comments have been addressed

2. Is the manuscript technically sound, and do the data support the conclusions?

Reviewer #2: Yes

3. Has the statistical analysis been performed appropriately and rigorously? 

Reviewer #2: Yes

4. Have the authors made all data underlying the findings in their manuscript fully available?

Reviewer #2: No

5. Is the manuscript presented in an intelligible fashion and written in standard English?

Reviewer #2: Yes

6. Review Comments to the Author

Reviewer #2: Metzger and colleagues have adequately addressed all of my comments. Indeed, the authors made a very good job; each question was clearly answered, and the authors have explained their methods (particularly, the statistical analyses used) in detail. Congratulations!

I have only two suggestions as follows.

Methods, line 113. Please report the data regarding ambient temperature in Celsius degree, as reported for the incubation temperature (line 198).

Methods, lines 366 and 367. Please replace “normality using R using the qq plot feature” with “normality using the qq plot feature of R software”.

7. PLOS authors have the option to publish the peer review history of their article (what does this mean?). If published, this will include your full peer review and any attached files.

Reviewer #2: Yes: Samuel Penna Wanner

---

## [Author Response · Author response to Decision Letter 1]

6 Dec 2019

The responses to reviewer and editor comments can be found in the uploaded word doc titled "Metzger_LetterResponsetoReviewersRoundTwo_ PLOS_12062019". The responses are also pasted below:

Please find in the following pages the referee comments quoted in italics. Our responses follow each one of the queries. 

“Reviewer #2: Metzger and colleagues have adequately addressed all of my comments. Indeed, the authors made a very good job; each question was clearly answered, and the authors have explained their methods (particularly, the statistical analyses used) in detail. Congratulations!”

<< We thank the reviewer for appreciating the updates we made to the manuscript and our responses to the reviewer’s questions. >>

“I have only two suggestions as follows.

Methods, line 113. Please report the data regarding ambient temperature in Celsius degree, as reported for the incubation temperature (line 198).”

<< Great catch! We have updated this line with the correct room temperature in degrees Celsius. >>

“Methods, lines 366 and 367. Please replace “normality using R using the qq plot feature” with “normality using the qq plot feature of R software”.”

<< We have updated the manuscript as suggested. >>

---

## [Editor Report · Decision Letter 2]

11 Dec 2019

Post mortem evaluation of inflammation, oxidative stress, and PPARγ activation in a nonhuman primate model of cardiac sympathetic neurodegeneration

PONE-D-19-17539R2

Dear Dr. Emborg,

We are pleased to inform you that your manuscript has been judged scientifically suitable for publication and will be formally accepted for publication once it complies with all outstanding technical requirements.

With kind regards,

Michael Bader

Academic Editor

PLOS ONE
---

## [Editor Report · Acceptance letter]

23 Dec 2019

PONE-D-19-17539R2 

Post mortem evaluation of inflammation, oxidative stress, and PPARγ activation in a nonhuman primate model of cardiac sympathetic neurodegeneration 

Dear Dr. Emborg:

I am pleased to inform you that your manuscript has been deemed suitable for publication in PLOS ONE. Congratulations! Your manuscript is now with our production department. 

With kind regards,

on behalf of

Prof. Michael Bader 

Academic Editor

PLOS ONE